# SceneTransporter: Optimal Transport-Guided Compositional Latent Diffusion for Single-Image Structured 3D Scene Generation

**Ling Wang**[1,2*]  **Haoxiang Guo**[3*]  **Xinzhou Wang**[2,4*]  **Fuchun Sun**[2†]  **Kai Sun**[2]
**Pengkun Liu**[2,5]  **Hang Xiao**[2,5]  **Zhong Wang**[1]  **Guangyuan Fu**[1]  **Eric Li**[3]
**Yang Liu**[3]  **Yikai Wang**[6†]
[1]Xi'an Research Institute of Hi-Tech  [2]Tsinghua University,  [3]SkyWork AI,
[4]Tongji University,  [5]Fudan University,  [6]Beijing Normal University
fcsun@mail.tsinghua.edu.cn  yikaiw@bnu.edu.cn

## Abstract

We introduce SceneTransporter, an end-to-end framework for structured 3D scene generation from a single image. While existing methods generate part-level 3D objects, they often fail to organize these parts into distinct instances in open-world scenes. Through a debiased clustering probe, we reveal a critical insight: this failure stems from the lack of structural constraints within the model's internal assignment mechanism. Based on this finding, we reframe the task of structured 3D scene generation as a global correlation assignment problem. To solve this, Scene-Transporter formulates and solves an entropic Optimal Transport (OT) objective within the denoising loop of the compositional DiT model. This formulation imposes two powerful structural constraints. First, the resulting transport plan gates cross-attention to enforce an exclusive, one-to-one routing of image patches to part-level 3D latents, preventing entanglement. Second, the competitive nature of the transport encourages the grouping of similar patches, a process that is further regularized by an edge-based cost, to form coherent objects and prevent fragmentation. Extensive experiments show that SceneTransporter outperforms existing methods on open-world scene generation, significantly improving instance-level coherence and geometric fidelity. Code and models will be publicly available at https://2019epwl.github.io/SceneTransporter/.

## 1 Introduction

The capacity to generate high-quality, scalable 3D scenes is a cornerstone for the next generation of immersive technologies and embodied AI. While advancements in generative AI promise scalable synthesis, the vast majority of scene generators still produce monolithic, unstructured meshes (Vodrahalli et al., 2024; Li et al., 2025; Xiang et al., 2025). For real-world pipelines, a fused 3D shell is functionally inert. Downstream tasks—including material assignment, realistic physics simulation, asset retrieval and placement, and fine-grained editing—require a structured scene mesh with explicit, instance-level object-context disentanglement. A common "divide and conquer" solution attempts this by first segmenting an input image, then generating a 3D model for each part, and finally assembling them into a scene (Huang et al., 2025; Chen et al., 2024b; Ardelean et al., 2024). This multi-stage pipeline, however, is inherently brittle; its heavy reliance on 2D segmentation makes it incapable of handling heavily occluded objects, and transforms even minor 2D segmentation flaws into significant 3D geometric artifacts.

In recent years, end-to-end structured generation has emerged as a promising alternative (Lin et al., 2025b; Tang et al., 2025; Chen et al., 2025a; Yang et al., 2025b), enabled by the development of large-scale repositories with explicit part annotations (e.g., Objaverse (Lin et al., 2025a)). In this paradigm, a scene is no longer represented as a single, indivisible latent code but as a collection of

---

*Equal contribution

†Corresponding authors

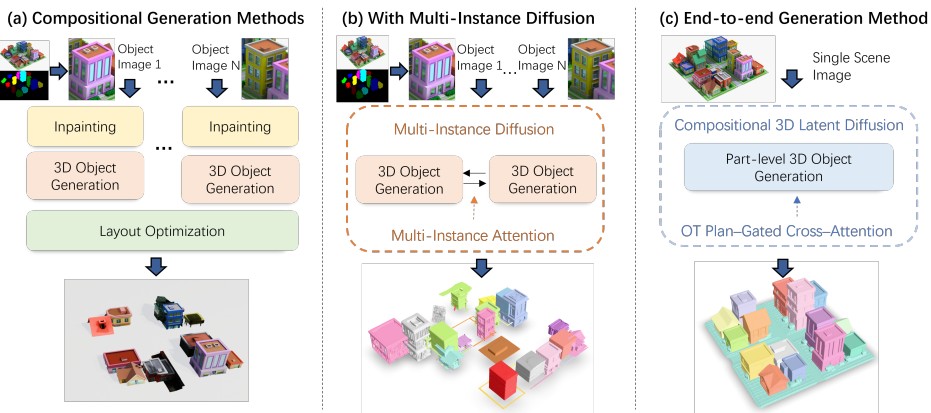

Figure 1: Comparison between our end-to-end scene generation pipeline in (c) with compositional 3D latent diffusion and existing "divide and conquer" methods.

disentangled latent tokens, where each set of tokens corresponds to a distinct 3D part. Although these methods show great promise for generating structured objects and indoor scenes, open-world structured scene generation remains underexplored. As illustrated in Figure 4, when naively applied to large, complex open-world scenes, it uncovers two persistent 3D pathologies: (i) **Structural Mispartition**—semantic instances within the scene fail to form disjoint parts; and (ii) **Geometric Redundancy**—multiple latents "compete" to describe the same geometric area.

To address these challenges, we first perform Probing Latent Structure with Debiased Clustering. This probe reveals the primary bottleneck: a lack of structural constraints within the model's assignment mechanism prevents the formation of stable instances. To this end, we introduce SceneTransporter, a framework that reframes the task as an Optimal-Transport–Guided Correlation Assignment problem. To solve this, we define a principled entropic Optimal Transport (OT) formulation that calculates a globally optimal transport plan between the set of image patch features and the part-level tokens. This optimization imposes powerful structural constraints through two key components. First, an OT Plan–Gated Cross–Attention module uses the plan to enforce a hard, one-to-one routing, ensuring each patch contributes to only one part, thus preventing feature entanglement. Second, the competitive nature of the transport incentivizes patches with high feature similarity to be assigned to the same token, naturally forming coherent structures. To further refine this grouping and ensure sharp object boundaries, we introduce an Edge–Regularized Assignment Cost, which penalizes assignments that cross salient image edges within the transport objective. Extensive experiments show that our approach outperforms existing methods, setting a new framework for structured 3D scene generation.

In summary, our contributions are as follows:

- We design a Debiased Clustering probe based on Canonical Correlation Analysis (CCA) to investigate the latent structure of part-level generators. The probe reveals that the core failure lies in the assignment mechanism due to a lack of structural constraints.

- We reframe the task as an Optimal-Transport (OT)–Guided Correlation Assignment problem. To solve this, we propose SceneTransporter, which leverages an entropic OT framework to impose two powerful structural constraints: an OT Plan–Gated Cross–Attention module for exclusive one-to-one routing, and an Edge-Regularized Assignment Cost for coherent structures grouping.

- SceneTransporter achieves state-of-the-art performance on open-world 3D scene generation, demonstrating significantly improved instance-level coherence and geometric fidelity.

## 2 RELATED WORK

**3D Scene Generation.** Early approaches to scene synthesis were dominated by retrieval-based methods, which compose scenes by aligning assets from a 3D database with an input image (Feng

et al., 2023; Gao et al., 2024; Langer et al., 2024). These methods, however, are fundamentally limited by the scope of their database and often suffer from alignment errors from a single view. The advent of large-scale generative models has shifted the focus towards creating novel content. One dominant approach is 2D-prior distillation, a multi-stage process that first generates consistent multi-view images (Chen et al., 2025b; Li et al., 2024), videos Sun et al. (2024); Wang et al. (2025); Liang et al. (2025), or panoramic images Yang et al. (2025a); Zhou et al. (2024) via powerful diffusion models, and then reconstructs a 3D scene from these views using techniques like Neural Radiance Fields (NeRF) or 3D Gaussian Splatting (3DGS). To address the geometric inconsistencies of 2D-lifting, another line of research develops models that operate directly in native 3D latent spaces (Meng et al., 2025; Wu et al., 2024; Lee et al., 2025; Ren et al., 2024; Lee et al., 2024; Liu et al., 2024b). While this improves multi-view consistency, it requires large-scale 3D datasets and generalizes poorly to in-the-wild scenes. Crucially, both of these generative avenues converge on the same limitation: their output is a single, unstructured monolithic mesh, which lacks the explicit object-level separation needed for most downstream tasks.

**3D Structured Scene Generation.** A common "divide and conquer" strategy for structured scene generation is to process a scene piece by piece (Chen et al., 2024b; Ardelean et al., 2024; Han et al., 2024). This typically involves a pipeline that segments the input, generates 3D models for the segments, and then arranges them. Such methods benefit from modularity but suffer from two key weaknesses: the accumulation of errors across stages and a failure to maintain global consistency. While more integrated approaches like MIDI (Huang et al., 2025) use a multi-instance attention mechanism to capture inter-object interactions, they remain fundamentally limited by their reliance on segmenting visible content, preventing the reconstruction of occluded parts. Another line of work explores end-to-end compositional generation using 3D diffusion models that bind latent token subsets to semantic parts (Lin et al., 2025b; Tang et al., 2025). While these models excel at generating structured objects, they fail to generalize to complex open-world scenes due to a scarcity of scene-level part annotations. This generalization gap manifests as two distinct geometric pathologies (Figure 4): structural mispartition and geometric redundancy. Our work offers the first direct solution, introducing a method to guide the compositional cross-attention of pretrained generators and explicitly correct these failure modes.

**Attention control in diffusion models.** Attention maps in image and video diffusion models have become a compact, training-free control interface: by manipulating or augmenting self- and cross-attention at sampling time, practitioners can steer layout and composition (Chen et al., 2024a; Hertz et al., 2022; Patashnik et al., 2023), transfer fine-grained texture and style (Hertz et al., 2024), enforce multi-prompt alignment (Chefer et al., 2023) and temporal coherence for videos (Cai et al., 2025; Liu et al., 2024a; Qi et al., 2023), and perform a variety of image-editing tasks (Parmar et al., 2023; Yang et al., 2023; Mokady et al., 2023; Park et al., 2024; Tumanyan et al., 2023). Despite its success in 2D, attention-based control has seen little application to 3D latent diffusion—largely because 3D representations differ fundamentally. In 2D, attention queries live on a regular spatial grid with explicit positions. By contrast, 3D data commonly takes irregular forms. For instance, in the vecset-based latent representation (Zhang et al., 2023) where a shape is encoded as an unordered set of latent vectors, positional information is not explicitly structured. The absence of a canonical spatial index therefore makes attention manipulation for 3D editing substantially more difficult and still largely unexplored.

## 3 METHODOLOGY

### 3.1 PRELIMINARIES: COMPOSITIONAL 3D GENERATORS

A typical compositional 3D generator consists of two key components: (i) a Transformer-based Variational Autoencoder (VAE), such as 3DShape2VecSet, which encodes a 3D mesh into a set of latent vectors and decodes them back into a geometric field (*e.g.*, SDF), and (ii) a denoising network that operates within this latent space to reverse a diffusion process and synthesize clean latents from noise. Prior to going further, we first introduce the notations and preliminaries used in our paper, especially those for the attention mechanism, as its core lies in modeling the geometric dependencies among shape components. At denoising step $t$, the model maintains $N$ part-specific token blocks $\{\mathbf{z}_i^{(t)}\}_{i=1}^N$, where each $\mathbf{z}_i^{(t)} \in \mathbb{R}^{K \times D}$ contains $K$ tokens (width $D$) for part $i$. A common practice is

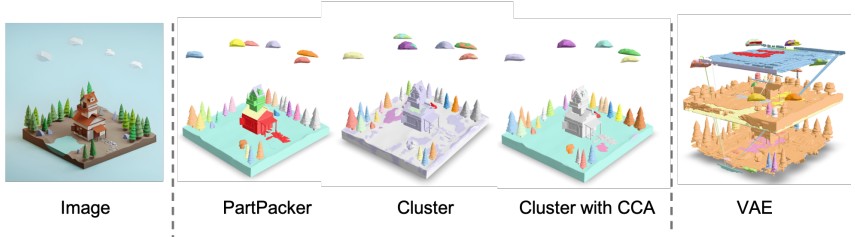

Figure 2: **Qualitative Results on Vecset-based Latent Probing.** *Cluster* and *Cluster with CCA* are our probes that perform in the compositional latent space of PartPacker; *VAE* clusters the latent obtained by encoding the fused geometry produced by PartPacker into the VAE. Colors denote part assignments.

to add a learnable *part identity* embedding $\mathbf{e}_i \in \mathbb{R}^D$ to all tokens of part $i$, where $\mathbf{1}_K$ is a column vector of ones:

$$\tilde{\mathbf{z}}_i^{(t)} = \mathbf{z}_i^{(t)} + \mathbf{1}_K \mathbf{e}_i^\top, \qquad \mathbf{Z}^{(t)} = \mathrm{concat}_{\mathrm{tokens}}\big(\tilde{\mathbf{z}}_1^{(t)}, \ldots, \tilde{\mathbf{z}}_N^{(t)}\big) \in \mathbb{R}^{(NK) \times D}. \tag{1}$$

Conditioning comes from a single RGB image $\mathbf{x}$ encoded by a frozen DINOv2 $\phi(\mathbf{x}) = \mathbf{I} \in \mathbb{R}^{L \times D_{\mathrm{img}}}$. Linear projections yield queries, keys, and values (omitting head indices for brevity):

$$\mathbf{Q} = \ell_Q(\mathbf{Z}^{(t)}) \in \mathbb{R}^{(NK) \times d}, \quad \mathbf{K} = \ell_K(\mathbf{I}) \in \mathbb{R}^{L \times d}, \quad \mathbf{V} = \ell_V(\mathbf{I}) \in \mathbb{R}^{L \times d}. \tag{2}$$

Row-normalized cross-attention fuses image evidence into the latent tokens. In practice, to increase expressiveness, *multi-head attention* is used: the projections are split into $H$ heads $(\mathbf{Q}_h, \mathbf{K}_h, \mathbf{V}_h)$, attention is computed per head $\mathbf{M}_h = \mathrm{Softmax}\big(\mathbf{Q}_h \mathbf{K}_h^\top / \sqrt{d}\big)$, the head outputs are concatenated, and a learned output projection produces the final update,

$$\hat{\mathbf{Z}}^{(t)} = \mathrm{Concat}_{h=1}^H\big(\mathbf{M}_h \mathbf{V}_h\big) \mathbf{W}_O. \tag{3}$$

The updated sequence proceeds through the DiT block and the denoising schedule. At inference, the final clean sequence is split back into $\{\hat{\mathbf{z}}_i\}_{i=1}^N$ and each subset is decoded by the pretrained VAE to obtain a part mesh; parts are then fused to form the full scene.

## 3.2 PROBING LATENT STRUCTURE WITH DEBIASED CLUSTERING

Part-level generators carry a strong inductive bias toward a part-level organization that models correlations among parts of the same object (e.g., chair legs and back share style and structure). In contrast, scene synthesis models an instance-level organization, where distinct entities are largely conditionally independent.

Applying the former to the latter gives rise to two observable failure modes (Figure 4): Structural Mispartition, where geometry for a single object is scattered across multiple part-tokens, and Geometric Redundancy, which leads to the geometry overlap between objects. Intriguingly, despite this incoherent part-level assignment, the union of all parts often reconstructs the overall scene reasonably well. This raises a concrete question: ***Can we recover a coherent, instance-level organization from these part-level latents?*** To probe this, we introduce a debiased clustering procedure with three steps: identify the shared subspace via canonical correlation analysis (CCA) on the part-level latent sets; suppress it by projecting tokens onto the orthogonal complement to isolate object-specific variation; and regroup the residual tokens to obtain semantically coherent sets. Implementation details are provided in Appendix B.

An illustrative example is provided in Figure 2. Our probe offers a clear signal: clustering raw part tokens fails to produce stable instance groupings, whereas clustering the CCA-debiased residual tokens reliably succeeds. This striking contrast reveals a critical flaw in current part-level generators: while their learned features implicitly contain the necessary information for correct associations, the models fail to establish these associations explicitly. As a result, the learned groupings are left weak, fragmented, and entangled with scene context. Based on this insight, we argue that a paradigm shift is required. Instead of hoping for an organization to emerge from an implicit learning process, we must introduce explicit structural constraints to guarantee a coherent, instance-level structure by design.

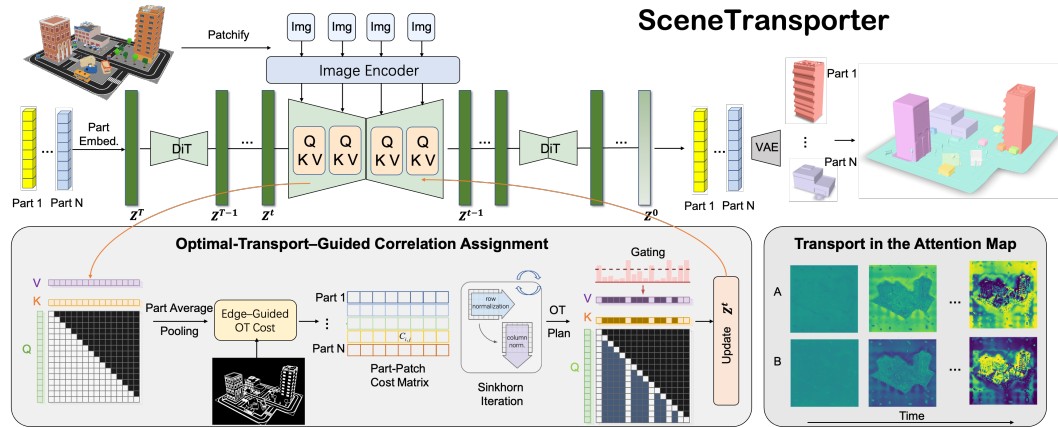

Figure 3: **Overview of the SceneTransporter pipeline.** At each denoising step $t$, our Optimal-Transport–Guided Correlation Assignment framework formulates a global OT problem between image patches and part-level tokens within the compositional latent DiT. We compute a part-patch cost from Q/K similarity, regularized by image edges, and solve for an optimal transport plan using Sinkhorn iteration. The OT plan gates the cross attention to enforce an explicit patch-to-part routing, and the resulting gated attention map updates the latent $z_t$. Attention maps transport over time, showing assignments becoming sharper and more instance-consistent.

## 3.3 OPTIMAL-TRANSPORT–GUIDED CORRELATION ASSIGNMENT

To impose the explicit structural constraints suggested by our probe, we reframe the routing of visual evidence to part tokens as a globally optimal correlation assignment problem. We propose to solve this problem at each denoising step using the principled framework of Optimal Transport (OT). OT provides the exact constraints needed to combat the failures we observed. By enforcing a one-to-one assignment, it ensures exclusivity, preventing the feature entanglement that causes objects to blend. Simultaneously, by requiring each part-latent to meet a coverage budget, it encourages the aggregation of large, coherent patch regions, directly mitigating semantic fragmentation. This OT-guided mechanism, which we detail next, transforms the generation from an unconstrained process into a structured, well-posed optimization.

**OT Problem Setup.** The core challenge of routing is to ensure a *globally consistent* allocation: image patches should not be double-counted by multiple 3D parts, and 3D parts should not compete for the same patch information. To address this, we model the assignment as a single optimization problem with constraints. We compute an entropic OT plan $\mathbf{A}_t \in \mathbb{R}^{N \times L}$ to allocate the mass from $L$ image patch features to the latent tokens of $N$ 3D parts:

$$\mathbf{A}_t = \arg\min_{\mathbf{A} \geq 0} \langle \mathbf{C}_t, \mathbf{A} \rangle + \varepsilon_t \, \mathcal{H}(\mathbf{A}) \quad \text{s.t.} \quad \mathbf{A}\mathbf{1} = \boldsymbol{\mu}, \ \mathbf{A}^\top \mathbf{1} = \boldsymbol{\nu}, \tag{4}$$

where the cost matrix $\mathbf{C}_t \in \mathbb{R}^{N \times L}$ measures the incompatibility between patches and parts, as will be detailed in Eq. (12). Crucially, the constraints enforce a budget for each part through $\boldsymbol{\mu} \in \Delta^N$, which prevents any part from being "starved" of information[1]. They also ensure that each patch contributes an equal amount of mass by $\boldsymbol{\nu} = \frac{1}{L}\mathbf{1}_L$, meaning each patch contributes $1/L$ of the total mass. $\mathcal{H}(\mathbf{A})$ is the entropy of the transport plan $\mathbf{A}$, and $\varepsilon_t$ is the entropic regularization term, which helps to smooth the solution. The resulting transport plan $\mathbf{A}_t$ minimizes the assignment cost while adhering to the capacity constraints.

**OT Plan–Gated Cross–Attention.** Given the entropic OT plan $\mathbf{A}_t \in \mathbb{R}^{N \times L}$ from Eq. (4), we inject this global assignment into the native cross-attention mechanism by using it to **gate** the incoming visual information multiplicatively. This approach enables us to regulate the amount of evidence each image patch can contribute to each 3D part, while preserving the standard $\mathrm{Softmax}$ attention to decide *which* of the available patches each token should focus on. To achieve this, we

---

[1]$\Delta^N = \{\boldsymbol{x} \in \mathbb{R}_{\geq 0}^N | \mathbf{1}_N \boldsymbol{x} = 1\}$ is a probability simplex. $\boldsymbol{\mu} \in \Delta^N$ satisfies $\sum_{i=1}^N \mu_i = 1$ with $\mu_i \geq 0$.

first convert the transport plan $\mathbf{A}_t$ into per-part patch weights $\boldsymbol{\omega}_i \in \Delta^L$ through row-normalization. We then design a bounded, identity-preserving gating function $\psi(\cdot)$ that transforms these weights into a gating signal:

$$\psi_{\lambda_t, \varepsilon_g}(w) \;=\; \varepsilon_g \;+\; (1 - \varepsilon_g)\, w^{\lambda_t}, \qquad w \in [0, 1],\; \lambda_t \geq 0,\; \varepsilon_g \in [0, 1), \tag{5}$$

where the guidance strength $\lambda_t$ determines how quickly the gate closes for patches with low weights, while a small floor $\varepsilon_g$ prevents any patch from being entirely starved of attention. Critically, when guidance is disabled ($\lambda_t = 0$), the function outputs exactly 1, guaranteeing that the mechanism reverts to standard cross-attention. This gating signal is subsequently applied to modulate both the keys and values in the attention computation for each part. For a specific head $h$, the keys $\mathbf{K}_h$ and values $\mathbf{V}_h$ are scaled row-wise by the gating signal derived from the part-specific weights $\omega_i(j)$. This results in a unique, gated "view" of the image memory for each part $i$:

$$\mathbf{K}_h^{(i)}(j, :) \;=\; \psi_{\lambda_t, \varepsilon_g}\big(\omega_i(j)\big)\, \mathbf{K}_h(j, :), \quad \mathbf{V}_h^{(i)}(j, :) \;=\; \psi_{\lambda_t, \varepsilon_g}\big(\omega_i(j)\big)\, \mathbf{V}_h(j, :). \tag{6}$$

The queries for the 3D part $i$ focus solely on this gated memory. By gating both keys and values, we ensure that routes suppressed by the OT plan contribute neither large logits nor large feature values, resulting in a clean, capacity-consistent routing with low leakage. The final representations for each part are computed via standard multi-head attention on their respective gated inputs and then stitched together:

$$\mathbf{M}_h^{(i)} = \mathrm{Softmax}\Big(\frac{\mathbf{Q}_h^{(t)}(\mathcal{S}_i, :)\,\big(\mathbf{K}_h^{(i)}\big)^\top}{\sqrt{d_h}}\Big), \qquad \mathbf{H}_h^{(i)} = \mathbf{M}_h^{(i)}\, \mathbf{V}_h^{(i)}, \tag{7}$$

$$\widehat{\mathbf{Z}}^{(t)}(\mathcal{S}_i, :) = \mathrm{Concat}_{h=1}^H\big[\mathbf{H}_h^{(i)}\big]\mathbf{W}_O, \quad i = 1, \dots, N, \qquad \widehat{\mathbf{Z}}^{(t)} \in \mathbb{R}^{(NK) \times d}. \tag{8}$$

**Edge–Regularized Assignment Cost.** In cluttered scenes, patch features near contact boundaries often seem compatible with multiple parts. As a result, the transported patch tends to "leak" information across adjacent objects. To address this issue, we introduce a weak but spatially precise prior: an *edge map* $\mathbf{E} \in [0, 1]^{H \times W}$ obtained from the conditioning image (*e.g.*, Canny/Sobel or a learned edge detector). The goal is to encourage *region–wise* consistency in the affinities between parts and patches while *discouraging* the spread of information across image edges.

Let $\{\bar{\mathbf{q}}_i^{(t)}\}_{i=1}^N$ represent the aggregated token of the $i$-th part. Meanwhile, let $\{\mathbf{k}_j\}_{j=1}^L$ denote the patch keys. We first compute the raw cosine similarities between these prototypes and keys,

$$S_{i,j} \;=\; \cos\big(\bar{\mathbf{q}}_i^{(t)}, \mathbf{k}_j\big) \in [-1, 1]. \tag{9}$$

We downsample the edge map to match the patch grid, resulting in $\mathbf{E}_\downarrow \in [0, 1]^{H_p \times W_p}$ where $H_p W_p = L$, and construct a 4–neighborhood graph among the patches. For a patch $j$ and its neighboring patch $\ell \in \mathcal{N}(j)$, we define an edge–aware coupling weight:

$$w_{j\ell} \;=\; \exp\Big(-\gamma_{\mathrm{edge}}\, \max\{\mathbf{E}_\downarrow(j), \mathbf{E}_\downarrow(\ell)\}\Big), \qquad \gamma_{\mathrm{edge}} > 0, \tag{10}$$

which is close to 1 in smooth regions and decays near image edges. We then perform a single edge–aware smoothing step on the affinities,

$$\widehat{S}_{i,j} \;=\; \frac{S_{i,j} \;+\; \lambda_{\mathrm{edge}}\sum_{\ell \in \mathcal{N}(j)} w_{j\ell}\, S_{i,\ell}}{1 \;+\; \lambda_{\mathrm{edge}}\sum_{\ell \in \mathcal{N}(j)} w_{j\ell}}, \qquad \lambda_{\mathrm{edge}} \geq 0. \tag{11}$$

Eq. (11) facilitate the spread of evidence within regions characterized by low edge strength, while simultaneously inhibiting the spread across edges. The process results in affinities that are piecewise smooth and respect boundaries. To further intensify the competition among parts for each patch, we implement a contrast normalization on a per-patch basis and obtain $\widetilde{S}_{i,j}$ from $\widehat{S}_{i,j}$.

Finally, our OT cost is defined as the margin–enhanced dissimilarity measure that incorporates the edge map to guide the assignment cost, ensuring that the affinities respect the boundaries and maintain region-wise consistency,

$$\mathbf{C}_t(i, j) \;=\; \frac{1}{2}\big(1 - \widetilde{S}_{i,j}\big). \tag{12}$$

| Method | Instance Mask | Geometry Fidelity | | | Part Disentanglement | | Inference Time (s) |
|---|---|---|---|---|---|---|---|
| | | ULIP↑ | ULIP-2↑ | Uni3D↑ | IoU$_{max}$ ↓ | IoU$_{mean}$ ↓ | |
| MIDIHuang et al. (2025) | ✓ | 0.1397 | 0.2763 | 0.2518 | 0.0458 | 0.1642 | 149.68 |
| PartCrafterLin et al. (2025b) | ✗ | 0.1177 | 0.3096 | 0.2635 | **0.0042** | **0.0539** | 157.97 |
| PartPackerTang et al. (2025) | ✗ | 0.1417 | 0.3083 | 0.2887 | 0.0319 | 0.2142 | **47.41** |
| Ours | ✗ | **0.1466** | **0.3220** | **0.3021** | 0.0101 | 0.0926 | 54.99 |

Table 1: **Quantitative Comparison on Structured 3D Scene Generation across Methods.** Bold values indicate the best scores, while underlined values indicate the second-best scores among the fair comparison.

When $\lambda_{edge} = 0$ (or the edge map consists entirely of zeros), Eq. (12) simplifies to the standard cosine–based cost with contrast normalization. A positive $\lambda_{edge}$ suppresses cross–boundary transport without requiring any semantic masks: patches on opposite sides of strong edges receive weak mutual support in Eq. (11), so the subsequent OT solver more reliably assigns them to different parts. All operations are differentiable and head/part–agnostic, and the hyperparameters $(\lambda_{edge}, \gamma_{edge})$ can be annealed across denoising steps (stronger in early steps, weaker later) to promote clean separation first and fine detail later.

# 4 EXPERIMENTS

## 4.1 EXPERIMENTAL SETUP

**Baselines** We compare our approach against recent state-of-the-art methods for part-level 3D generation—PARTCRAFTER (Lin et al., 2025b), PARTPACKER (Tang et al., 2025), and MIDI (Huang et al., 2025). We ensure fairness in our evaluation by using the officially released source code and checkpoints for each baseline.

**Metrics.** Following prior work (Tang et al., 2025; Zhao et al., 2025), we evaluate geometry fidelity with ULIP (Xue et al., 2023; 2024) and Uni3D (Zhou et al., 2023). These models learn unified representations across text, image, and point cloud modalities. Since both ULIP-2 and Uni3D require colored point clouds as input, we assign a uniform white color to all mesh outputs before computing the metrics. For part disentanglement, we voxelize the canonical space into a $64^3$ grid, binarize occupancies $\{\mathbf{O}_i\}_{i=1}^{N}$ for the $N$ generated parts, and compute pairwise Intersection-over-Union (IoU). We report the mean of the top-20 largest IoUs, and the maximum IoU; lower values indicate better disentanglement (less inter-part overlap).

**Implementation details.** We build *SceneTransporter* on the open-source part-level 3D generator of Tang et al. (2025), which uses a rectified-flow DiT with 24 attention blocks and supports arbitrary part counts via a dual-volume packing strategy. In our setup we instantiate dual volumes, yielding a compositional latent of size $4096 \times 2 = 8192$ with channel width 64. For the OT solver, we use a stabilized log-domain Sinkhorn with 40 iterations. We enable OT plan–gated attention in the first half of the DiT blocks, and use standard cross-attention in the remaining blocks to refine the global geometry. All other inference settings follow Tang et al. (2025).

## 4.2 COMPARISONS WITH STATE-OF-THE-ARTS.

To demonstrate the effectiveness and breadth of our approach, we evaluate on an open-world set of 74 high-quality scene images collected from the Web, spanning diverse styles. As shown in Table 1, our method achieves the highest geometry fidelity and the second-lowest inter-part overlap. The absolute lowest IoU is reported by PartCrafter, largely because it discards background/ground regions during generation, which trivially reduces overlap but also compromises scene completeness. Although our runtime is slightly slower than PartPacker, we deliver substantially better geometry and disentanglement, while remaining much faster than MIDI and PartCrafter.

Figure 4 provides qualitative comparisons: our method produces coherent object-level parts (e.g., complete houses, sofas, trees, lamps), whereas PartPacker shows semantic fragmentation (e.g., roofs or tree canopies split across parts) and feature entanglement (e.g., ground features leaking into adja-

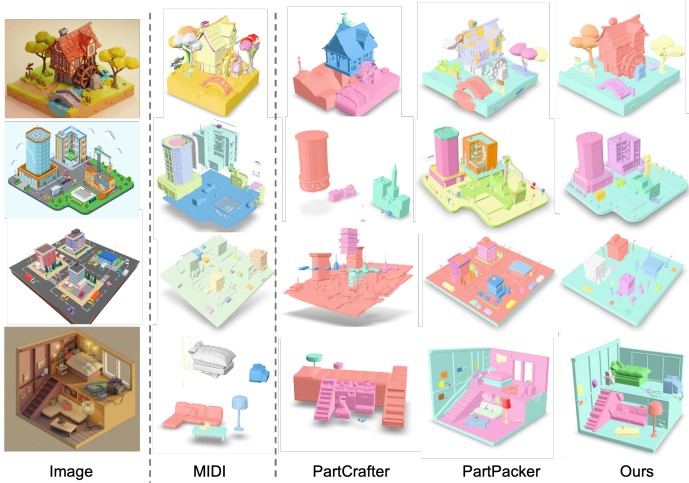

Figure 4: **Qualitative Comparison on Structured 3D Scene Generation across Methods.** Different colors indicate different parts in the generated 3D scene.

| Method | Geometry ↑ | Layout ↑ | Segmentation ↑ |
|---|---|---|---|
| MIDI (Huang et al., 2025) | 2.61 | 1.82 | 2.29 |
| PartCrafter (Lin et al., 2025b) | 2.44 | 1.63 | 2.17 |
| PartPacker (Tang et al., 2025) | 2.81 | 2.95 | 1.97 |
| Ours | **3.09** | **3.34** | **3.22** |

Table 2: **User Study.** Human evaluation of different structure 3D scene generation methods across multiple aspects. Scores range from 1 to 4, with higher scores indicating better performance. Bold values represent the best performance within each metric.

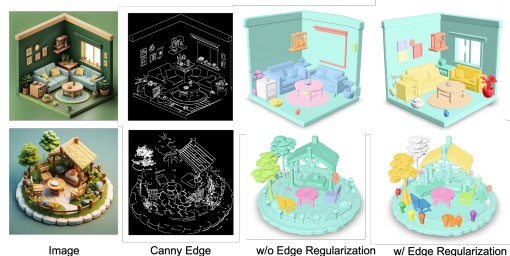

Figure 5: **Qualitative Ablation Studies on the Edge–Regularized Assignment Cost.**

cent buildings). Trained primarily on indoor assets, MIDI further requires additional instance masks at inference; it performs reasonably on simple indoor scenes but degrades on outdoor cases, exhibiting spatial layout distortions and weaker instance separation. PartPacker does not use masks and can perform well on objects that are well separated from their surroundings, but its performance deteriorates in complex spatial layouts.

We invite 30 participants to evaluate three baselines and our method, considering three criteria: geometry quality, layout coherence, and segmentation plausibility. We employ a Forced Ranking Scale, where items are ranked from 1 to 4, with the highest rank receiving a score of 4 and the lowest rank receiving a score of 1. As clearly indicated in Table 2, our method receives the highest preference across all three criteria, indicating more coherent object–level parts, reduced feature leakage, and better scene–wide layout consistency.

## 4.3 ABLATION STUDY

**Effects of OT Plan–Gated Cross–Attention** As shown in Figure 6 (a), our OT Plan–Gated Cross–Attention method produces highly structured and focused attention maps. Notice how A_atten. map clearly isolates the ground, while the other (B_atten. map) concentrates exclusively on the houses. This clean separation of duties, visualized in the hard affinity map, results in distinct, non-overlapping regions. Consequently, each part is generated as a complete and clean geometric object (A_geo. for the ground, B_geo. for the houses). When combined, they form a perfectly organized scene Uni_geo. with sharp boundaries. In contrast, the standard cross-attention in (b) is noisy and chaotic. The attention maps are diffuse, sending mixed signals about which part is responsible for which region. This confusion leads to corrupted geometry. This result validates the efficacy of our OT Plan–Gated Cross–Attention module, proving that its enforcement of one-to-one constraints effectively prevents feature entanglement.

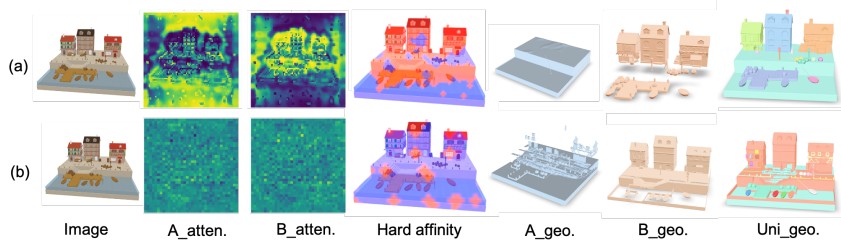

Figure 6: **Qualitative Ablation Studies on the OT Plane-gated Cross Attention.** Here, A_attn. and B_attn. denote the dual-volume soft attention probability maps, reshaped to the image patch grid (brighter means higher affinity). Hard affinity visualizes the argmax(A,B) patch assignments overlaid on the input image (blue→A, red→B). A_geo. and B_geo. are the geometries decoded from dual volumes, respectively, and Uni_geo. is their fused scene mesh. Row (a) shows our OT plan–gated cross-attention; row (b) shows the standard cross-attention.

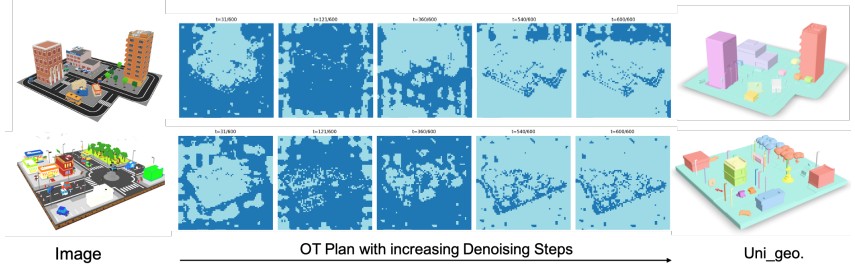

Figure 7: **Qualitative Ablation Studies on the OT Plan Progression over Denoising Steps.** Each map visualizes the hard OT plan at a given denoising step: every cell is an image patch assigned to one volume (dark blue = A, light cyan = B). Left→right shows the OT plan's evolution; later steps mostly stabilize with only local refinements.

**OT Plan Progression over Denoising Steps.** As shown in Figure 7, as $t$ increases, the OT plan quickly stabilizes: after roughly $t \approx 540/600$ the global partition changes little. In the late denoising stage, entire objects (e.g., buildings, furniture, trees) are already routed into a single volume, with only fine adjustments thereafter. This temporal behavior explains the coherence of our object-level parts: coarse, semantic routing is decided early and preserved, while later steps polish details without flipping the global assignment.

**Effects of Edge–Regularized Assignment Cost.** Our correlation assignment operates at the patch level. Because image features are locally smooth, neighboring patches usually prefer the same part, which tends to pull an entire connected region into a single part. While desirable within an object, this behavior can mistakenly merge *spatially adjacent but semantically distinct* objects at contact zones (e.g., furniture touching walls or fences touching posts). As shown in Figure 5, adding the edge regularizer cleanly separates objects that are contiguous in the image—the sofa from the corner side table in the top row, and the wooden posts from the surrounding fence in the bottom row. Compared to the version without edge regularization, the edge-aware plan yields crisper inter-object boundaries, fewer mixed parts, and improved structural fidelity, while requiring no additional instance mask supervision.

## 5 CONCLUSION

In this paper, we introduced SceneTransporter, a novel framework for structured 3D scene generation from a single image. By reframing the task as a global correlation assignment problem and solving it with an Optimal Transport layer, our method imposes powerful structural constraints directly on the generative process, effectively resolving the critical issues of structural mispartition and geometric redundancy found in existing models. Experimental results demonstrate that our method achieves state-of-the-art performance, generating complex open-world scenes with significantly improved geometric fidelity and instance-level coherence.

ACKNOWLEDGMENTS

This work is supported by the National Science and Technology Major Project of the Ministry of Science and Technology of China (No. 2025ZD1206301), the National Natural Science Foundation of China (No. 62576043), and the Natural Science Foundation of Beijing, China (No. L233006).

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

## A  THE USE OF LARGE LANGUAGE MODELS

The writing of this paper was assisted by Large Language Models (LLM). Specifically, the LLM was utilized for the following tasks:

- Improving grammar, clarity, and academic tone throughout the manuscript.
- Rephrasing and restructuring sentences and paragraphs to enhance the logical flow of our arguments.

In accordance with ICLR policy, the human authors directed all content generation, critically reviewed and edited all model outputs, and take full and final responsibility for the claims, accuracy, and integrity of this work.

## B  METHOD DETAILS

**Debiased Clustering Probe**  To quantitatively investigate the flawed latent organization of the baseline, we designed a diagnostic probe based on unsupervised clustering. The key idea is that the constituent information for complete objects is indeed present within the generated latents, but is merely disorganized and entangled by the model's component-based prior. Specifically, let $\mathbf{Z}_A, \mathbf{Z}_B \in \mathbb{R}^{K \times D}$ be the dual-volume latents generated by Tang et al. (2025) for a given scene. We stack them and apply a mild whitening:

$$\mathbf{Z} = \begin{bmatrix} \mathbf{Z}_A \\ \mathbf{Z}_B \end{bmatrix}, \qquad \tilde{\mathbf{Z}} = (\mathbf{Z} - \mathbf{1}\mu^\top)\Sigma^{-1/2}, \quad \mu = \frac{1}{2K}\sum_{i=1}^{2K}\mathbf{Z}_i, \ \ \Sigma = \frac{1}{2K}\sum_{i=1}^{2K}(\mathbf{Z}_i - \mu)(\mathbf{Z}_i - \mu)^\top, \ \ (13)$$

where $\Sigma^{-1/2}$ is computed from an eigendecomposition with a small ridge for stability. Denote the whitened halves by $\tilde{\mathbf{Z}}_A, \tilde{\mathbf{Z}}_B$.

Directly clustering the raw token latents often fails because the process is dominated by strong, shared nuisance factors—such as the ground plane or global style—that are non-diagnostic for object identity. To suppress these pervasive cross-volume trends, we first estimate a shared subspace between the two latent volumes $(\tilde{\mathbf{Z}}_A, \tilde{\mathbf{Z}}_B)$ using canonical correlation analysis (CCA). CCA identifies paired directions $(\mathbf{u}_j, \mathbf{v}_j)$ that maximize the correlation between the volumes' projections. We retain all directions whose canonical correlation $\rho_j$ exceeds a threshold $\tau$ to define this shared subspace

$$\{(\rho_j, \mathbf{u}_j, \mathbf{v}_j)\}_{j \in \mathcal{J}} = \text{CCA}(\tilde{\mathbf{Z}}_A, \tilde{\mathbf{Z}}_B), \quad \mathcal{J} = \{j : \rho_j > \tau\}, \quad \mathcal{U}_{\text{shared}} = \text{span}\{\mathbf{u}_j, \mathbf{v}_j : j \in \mathcal{J}\}, \tag{14}$$

where $\text{span}\{\cdot\}$ denotes the set of all linear combinations of the listed vectors. Let $\mathbf{U}$ be a column-orthonormal basis of $\mathcal{U}_{\text{shared}}$ and $\mathbf{P} = \mathbf{U}\mathbf{U}^\top$ the orthogonal projector. We obtain *debiased* tokens by removing their shared component:

$$\hat{\mathbf{Z}}_A = \tilde{\mathbf{Z}}_A - \tilde{\mathbf{Z}}_A\mathbf{P}, \qquad \hat{\mathbf{Z}}_B = \tilde{\mathbf{Z}}_B - \tilde{\mathbf{Z}}_B\mathbf{P}. \tag{15}$$

Intuitively, (15) down-weights the high-variance global modes while preserving object-specific variation. We then cluster the debiased tokens $\hat{\mathbf{Z}} = [\hat{\mathbf{Z}}_A; \hat{\mathbf{Z}}_B]$ with a flexible Gaussian mixture

$$\hat{\mathbf{Z}} \sim \sum_{c=1}^{C} \pi_c \mathcal{N}(\boldsymbol{\mu}_c, \boldsymbol{\Sigma}_c), \qquad C = 2, \tag{16}$$

and denote by $\gamma_{ic}$ the posterior responsibility of component $c$ for token $i$. To improve robustness, tokens with low maximum confidence ($\max_c \gamma_{ic} < \delta$) are reassigned to the nearest centroid computed from high-confidence members ($\max_c \gamma_{ic} \geq \delta$). Finally, grouped tokens are decoded independently with the frozen VAE decoder to visualize the resulting object-level organization.

## C  EXPERIMENT SETTINGS

### C.1  HYPERPARAMETERS

Table 3 provides the hyperparameters needed to replicate our experiments.

Table 3: Hyperparameters of OT plan–gated cross-attention used in all experiments.

| Symbol | Description | Value |
|--------|-------------|-------|
| $\varepsilon_t$ | Entropic regularization weight | 0.10 |
| $\lambda_{\text{edge}}$ | Edge regularization strength | 0.8 |
| $\gamma_{\text{edge}}$ | Edge sensitivity | 8.0 |
| $\lambda_t$ | Guidance strength | 2.5 |
| $\varepsilon_g$ | Floor term | 0.02 |
| $K_{\text{OT}}$ | Number of Sinkhorn iterations | 40 |

## C.2 HUMAN EVALUATION.

In our user study, we compare our method with three baselines (MIDI (Huang et al., 2025), PartCrafter (Lin et al., 2025b), and PartPacker (Tang et al., 2025)) on perceptual quality. For each reference image, all four methods generate a structured 3D scene, which we export as `.glb` meshes. These meshes are loaded into a web-based 3D viewer with identical lighting and shading settings. The four methods are assigned to labels A–D in a single random permutation, and the corresponding meshes are displayed side-by-side under these labels. Participants can freely rotate, zoom, and inspect each mesh interactively.

We recruit a total of **30 participants** (graduate students and researchers in computer vision/graphics not involved in this project). For each reference image they are shown, participants evaluate the four methods along three dimensions. Specifically, they are asked to *rank* the four scenes (A–D) from 1 (lowest) to 4 (highest) for each of the following questions:

- **Geometry Quality:** "*Please rank the overall geometry quality of each scene.*" This metric evaluates how detailed, precise, and faithful to the reference image the geometry of each scene is, including fine-grained structures and overall shape fidelity.

- **Layout Coherence:** "*Please rank the overall spatial layout and arrangement of objects in each scene relative to the reference image.*" This measures how coherent the scene layout is, and how well object positions, scales, and composition align with the input image.

- **Segmentation Plausibility:** "*Please rank the overall plausibility of the object-level part decomposition in each scene.*" This assesses to what extent each scene exhibits clear, reasonable instances with minimal overlaps, missing regions, or mixed parts.

For each method and each criterion, we compute the *average rank* over all evaluated images and participants. These averaged scores are reported in Table 2 of the main paper, where higher values indicate stronger human preference. Our method achieves the highest average rank across all three dimensions, suggesting more coherent object-level parts, reduced feature leakage, and better scene-wide layout consistency.

## D ADDITIONAL EXPERIMENTS

### D.1 REAL-WORLD RESULTS

In our current setting, all methods (including the baselines) are trained on synthetically rendered scenes, which allows us to construct a large and diverse training set with consistent part-level annotations. To test the generalization ability of our model to natural photographs, we perform *zero-shot* evaluation on real-world images from the DL3DV-10K dataset (Ling et al., 2024).

As expected, directly applying the synthetic-trained model to raw photographs leads to a noticeable performance drop due to the appearance domain gap. Following the strategy proposed in PartCrafter (Lin et al., 2025b), we therefore explore transferring the style of real-world images to make them look more like images rendered from a graphics engine using recent image editing models, such as GPT-5. Specifically, we use prompts of the form: "*Preserve all details and perform image-to-image style transfer to convert the image into the style of a 3D rendering (Objaverse-style rendering).*" The style-transferred images preserve the original scene layout and object identities, while matching the rendering statistics of our synthetic training data.

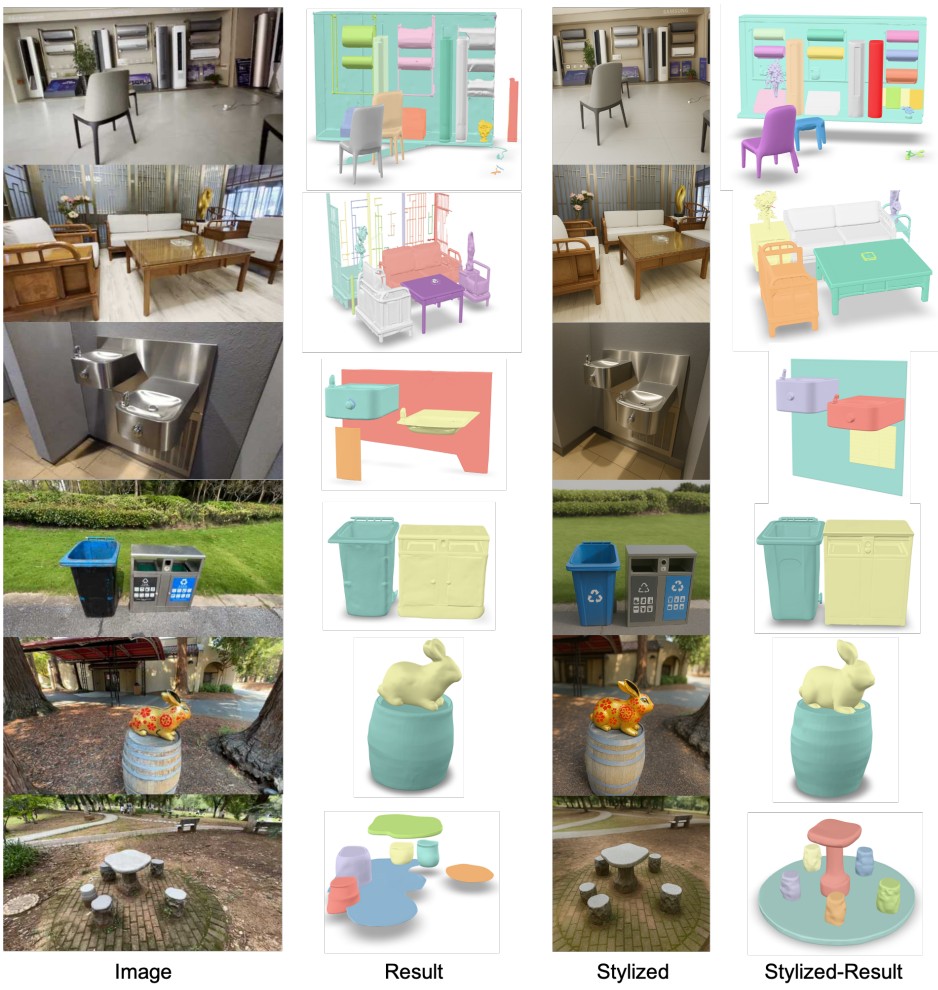

|  Image | Result | Stylized | Stylized-Result |

Figure 8: **Qualitative Results on Structured 3D Scene Generation from Real-World Images.**
We use a GPT-5–based image editing model to transfer the style of real-world images, making them
look like images rendered from a graphics engine.

Under this simple, purely test-time preprocessing, our method produces significantly more faithful
and coherent 3D scene reconstructions: objects are better separated, part boundaries align more
closely with image evidence, and the overall layout matches the input photograph more accurately.
As shown in Figure 8, this simple strategy works surprisingly well on a variety of indoor and outdoor
real-world scenes.

### D.2 CONVERGENCE ANALYSIS OF THE ENTROPIC OT SOLVER

In each OT-gated cross-attention layer, we solve a small entropic optimal transport (OT) problem
between two "rolls" of part queries and $M$ image tokens. Given a cost matrix $C \in \mathbb{R}^{2 \times M}$, row
marginals $\mu \in \mathbb{R}^2$ and column marginals $\nu \in \mathbb{R}^M$, we minimize

$$\min_{A \geq 0} \langle C, A \rangle + \varepsilon H(A) \quad \text{s.t.} \quad A\mathbf{1} = \mu, \quad A^\top \mathbf{1} = \nu, \tag{17}$$

where $A \in \mathbb{R}^{2 \times M}$ is the transport plan, $\varepsilon > 0$ is the entropic regularization strength, and $H(A) = \sum_{ij} A_{ij} \log A_{ij}$ is the negative-entropy regularizer.

Following the standard dual formulation of entropic OT, we introduce dual potentials $f \in \mathbb{R}^2$ (for
the row constraints) and $g \in \mathbb{R}^M$ (for the column constraints). At Sinkhorn iteration $k$, the corre-

sponding transport plan has the form

$$A_{ij}^{(k)} \;\propto\; \exp\!\left(\frac{f_i^{(k)} + g_j^{(k)} - C_{ij}}{\varepsilon}\right), \tag{18}$$

followed by a normalization step to enforce the marginal constraints $A^{(k)}\mathbf{1} \approx \mu$ and $(A^{(k)})^\top \mathbf{1} \approx \nu$. A fixed point of the Sinkhorn updates corresponds to a stationary solution of the entropic OT problem equation 17.

**Monitored residuals.** To assess optimization stability and convergence in practice, we instrument our entropic OT solver and log convergence statistics for all OT-gated cross-attention layers. For each Sinkhorn solve, at each iteration $k$ we record the following residuals:

- **Dual updates**: the $\ell_2$ norms of the changes in the dual variables

$$r_f^{(k)} = \left\| f^{(k+1)} - f^{(k)} \right\|_2, \qquad r_g^{(k)} = \left\| g^{(k+1)} - g^{(k)} \right\|_2. \tag{19}$$

  These measure how much the dual potentials still move between two successive iterations.

- **Transport-plan update**: the Frobenius norm of the change in the transport plan

$$r_A^{(k)} = \left\| A^{(k+1)} - A^{(k)} \right\|_F, \tag{20}$$

  which quantifies how much the routing plan is still being updated.

- **Marginal-constraint violation**: the deviation of the current plan from the prescribed marginals,

$$r_{\text{row}}^{(k)} = \frac{1}{2}\sum_{i=1}^{2}\left|(A^{(k)}\mathbf{1})_i - \mu_i\right|, \qquad r_{\text{col}}^{(k)} = \frac{1}{M}\sum_{j=1}^{M}\left|((A^{(k)})^\top\mathbf{1})_j - \nu_j\right|, \tag{21}$$

  and we report their sum

$$r_{\text{marg}}^{(k)} = r_{\text{row}}^{(k)} + r_{\text{col}}^{(k)}. \tag{22}$$

  Intuitively, $r_{\text{marg}}^{(k)}$ measures how well the current $A^{(k)}$ satisfies the OT marginal constraints.

In words, $r_f^{(k)}, r_g^{(k)}$ and $r_A^{(k)}$ tell us whether the dual variables and the transport plan have stabilized, while $r_{\text{marg}}^{(k)}$ indicates how strictly the mass conservation constraints are enforced. Figure. 9 visualizes these quantities for three OT-gated cross-attention layers. For each layer, we run the Sinkhorn solver with a fixed number of iterations (40 in all our experiments) and plot the residuals $r_f^{(k)}, r_g^{(k)}, r_A^{(k)}$ and $r_{\text{marg}}^{(k)}$ as a function of the iteration index $k$.

Across all OT-gated layers and across different denoising steps, we observe a consistent pattern: the dual and plan residuals $r_f^{(k)}, r_g^{(k)}$, and $r_A^{(k)}$ drop by 3–5 orders of magnitude within only 3–5 Sinkhorn iterations and then remain numerically flat, while the marginal-constraint violation $r_{\text{marg}}^{(k)}$ quickly converges to a very small value and stays stable without oscillation or divergence. This indicates that the entropic OT subproblem in our setting is well-conditioned and that our solver converges rapidly and stably under the default choice of 40 iterations used in all experiments.

### D.3 Additional Ablation Studies

Table 4 presents quantitative ablations regarding the core components and hyperparameters of our OT-guided routing mechanism. The key observations are summarized below.

**Impact of OT Plan–Gated Attention and Edge-Regularized Cost.** Rows (a.1)–(a.3) isolate the contributions of the proposed OT modules. The removal of the OT Plan–Gated Cross-Attention (a.1) precipitates a discernible drop in Geometry Fidelity (e.g., ULIP decrease) and a marked deterioration in IOU metrics. This empirical evidence substantiates our hypothesis that enforcing a one-to-one, capacity-constrained patch-to-part routing is critical for mitigating feature leakage and suppressing redundant geometry. Furthermore, while retaining OT gating but omitting the Edge-Regularized

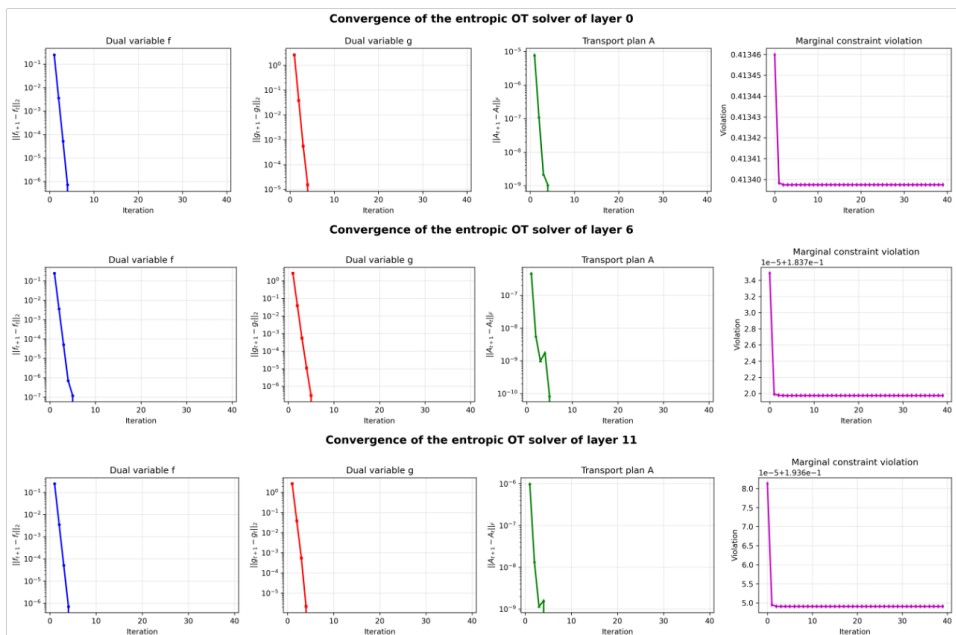

Figure 9: **Convergence of the entropic OT solver across OT-gated cross-attention layers.** We plot the residuals of the dual variables and transport plan, as well as the marginal-constraint violation, for three representative OT-gated cross-attention layers.

Assignment Cost (a.2) improves upon the baseline, the IoU metrics remain suboptimal compared to the full configuration. The full model (a.3) attains superior performance across all metrics, confirming that edge-aware smoothing effectively refines object separation without compromising global geometric fidelity. In terms of efficiency, the inclusion of OT modules introduces a manageable computational overhead (increasing inference time from 47.4,s to 55.0,s), while memory consumption remains negligible.

**Effect of OT Hyperparameters.** We further analyze the sensitivity of our method to key hyperparameters in Rows (b)–(d). First, varying the entropic regularization $\varepsilon_t$ (Rows b.1–b.4) reveals that $\varepsilon_t = 0.10$ yields the optimal trade-off. Deviating to lower or higher values disrupts the balance between transport plan sparsity and smoothness, leading to a degradation in both geometric fidelity (ULIP/Uni3D) and part disentanglement metrics. Second, regarding the edge sensitivity $\gamma_{\text{edge}}$ (Rows c.1–c.4), a lower value (6.0) relaxes boundary constraints, which marginally benefits global geometry (highest ULIP) but causes leakage across object boundaries (increased $\text{IoU}_{\text{max}}$). Conversely, excessive sensitivity ($\geq 10.0$) over-constrains the routing, harming all metrics. Our default $\gamma_{\text{edge}} = 8.0$ strikes the best balance. Finally, for the edge-smoothing weight $\lambda_{\text{edge}}$ (Rows d.1–d.3), we observe that sufficient smoothing is required to enforce intra-part coherence. Setting $\lambda_{\text{edge}}$ too low (0.6) or too high (1.0) results in suboptimal segmentation, evident from the sharp rise in $\text{IoU}_{\text{max}}$. The default $\lambda_{\text{edge}} = 0.8$ consistently achieves superior performance, demonstrating that the method is robust within a reasonable range around the optimal settings.

**Number of OT-Gated DiT Blocks.** Finally, Rows (e.1)–(e.3) investigate the optimal density of OT Plan–Gated Cross-Attention within the DiT architecture. Restricting OT integration to only one-third of the blocks (e.1) yields some geometric improvement over the baseline but proves insufficient for effective part separation, as evidenced by suboptimal IoU metrics. Increasing the coverage to half of the blocks (e.2, our default) precipitates a substantial gain in both Geometry Fidelity and Part Disentanglement. Crucially, this setting maintains a moderate runtime, incurring only a reasonable overhead compared to the baseline. Applying OT to all blocks (e.3) provides marginal gains in specific geometry metrics (e.g., ULIP-2) but leads to diminishing returns—or even slight degradation—in part metrics, while imposing a significant latency penalty. These results indicate that integrating OT into approximately half of the layers offers the most favorable trade-off between structural disentanglement and computational efficiency.

| Setting | Geometry Fidelity | | | Part Disentanglement | | Inference Time (s) | Inference Memory (M) |
|---|---|---|---|---|---|---|---|
| | ULIP↑ | ULIP-2↑ | Uni3D↑ | IoU$_{max}$ ↓ | IoU$_{mean}$ ↓ | | |
| (a.1) w/o OT Plan–Gated Cross–Attention | 0.1417 | 0.3083 | 0.2887 | 0.0319 | 0.2142 | 47.41 | 9030 |
| (a.2) w/o Edge–Regularized Assignment Cost | 0.1452 | 0.3164 | 0.2916 | 0.0241 | 0.1136 | 54.61 | 9068 |
| (a.3) Full model | **0.1466** | **0.3220** | **0.3021** | **0.0101** | **0.0926** | 54.99 | 9070 |
| (b.1) $\varepsilon_t = 0.08$ | 0.1460 | 0.3184 | 0.3003 | 0.0117 | **0.0914** | 55.30 | 9070 |
| (b.2) $\varepsilon_t = 0.10^*$ | **0.1466** | **0.3220** | **0.3021** | **0.0101** | 0.0926 | 54.99 | 9070 |
| (b.3) $\varepsilon_t = 0.12$ | 0.1437 | 0.3185 | 0.2970 | 0.1239 | 0.1119 | 55.12 | 9070 |
| (b.4) $\varepsilon_t = 0.14$ | 0.1424 | 0.3178 | 0.2937 | 0.0159 | 0.0936 | 55.02 | 9070 |
| (c.1) $\gamma_{edge} = 6.0$ | **0.1488** | **0.3239** | 0.3014 | 0.0243 | 0.0936 | 56.17 | 9070 |
| (c.2) $\gamma_{edge} = 8.0^*$ | 0.1466 | 0.3220 | 0.3021 | **0.0101** | **0.0926** | 54.99 | 9070 |
| (c.3) $\gamma_{edge} = 10.0$ | 0.1457 | 0.3182 | **0.3029** | 0.0570 | 0.1113 | 55.17 | 9070 |
| (c.4) $\gamma_{edge} = 12.0$ | 0.1426 | 0.3156 | 0.3003 | 0.1149 | 0.0994 | 55.89 | 9070 |
| (d.1) $\lambda_{edge} = 0.6$ | 0.1456 | **0.3236** | 0.3003 | 0.0609 | 0.1128 | 53.59 | 9070 |
| (d.2) $\lambda_{edge} = 0.8^*$ | **0.1466** | 0.3220 | **0.3021** | **0.0101** | **0.0926** | 53.97 | 9070 |
| (d.3) $\lambda_{edge} = 1.0$ | 0.1439 | 0.3204 | 0.3017 | 0.1482 | 0.0952 | 55.17 | 9070 |
| (e.1) w 1/3 DiT blocks | 0.1465 | 0.3170 | 0.3004 | 0.0992 | 0.1061 | 51.89 | 9070 |
| (e.2) w 1/2 DiT blocks$^*$ | **0.1466** | 0.3220 | 0.3021 | **0.0101** | **0.0926** | 54.99 | 9070 |
| (e.3) w all DiT blocks | 0.1426 | **0.3262** | **0.3022** | 0.0207 | 0.0830 | 65.24 | 9070 |

Table 4: **Comparison of metrics for ablation.** Bold values indicate the best scores, while underlined values indicate the second-best scores among the fair comparison. Asterisk (∗) indicates the default settings in our method.

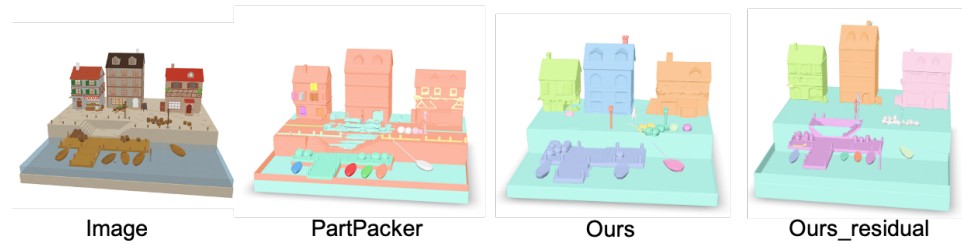

Image     PartPacker     Ours     Ours_residual

Figure 10: **Effect of adding a residual vanilla cross-attention branch.** Left to right: input image, PartPacker baseline, our OT-guided routing, and our OT-guided routing with an additional residual cross-attention branch (Ours_residual), which recovers the small crowded instances (e.g., boats) while preserving the improved global layout and part separation.

## D.4 FAILURE CASES AND RESIDUAL DIAGNOSTIC

**Crowded tiny instances.** In scenes with very dense clusters of small, similar objects that share only a few image patches (e.g., tightly packed boats or trees), the OT-guided routing can allocate most capacity to the strongest responses and effectively merge a few weak instances into their neighbours. This may slightly under-count very small repeated objects, while the global layout (buildings, roads, docks, etc.) remains correct. We find that adding a small residual branch of vanilla cross-attention on top of the OT-gated attention largely mitigates this issue, keeping the OT plan as a low-frequency structural prior and using the residual attention to recover high-frequency local details. Figure 10 visualizes a representative example, comparing (i) the baseline, (ii) SceneTransporter with pure OT-guided routing, and (iii) SceneTransporter with the residual cross-attention branch: the residual variant recovers the missing tiny instances while preserving the cleaner global structure of OT.

**Geometry artifacts.** We occasionally see nonsmooth surfaces or floating parts when the denoising schedule is too short or the number of tokens is too small. Increasing the number of diffusion steps or the token budget alleviates these cases, and the resulting artifacts are typically local and do not affect the overall scene layout.

**Out-of-distribution appearance.** Since the model is trained on synthetic rendered images, strongly out-of-distribution real images (e.g., unusual lighting or textures) can lead to degraded geometry and part grouping. As discussed in Appendix D.1, applying automated style transfer to convert real images into a 3D-rendering style significantly improves the results, and SceneTransporter still produces plausible structured scenes on many challenging real-world inputs.

