# OpenReview forum: "SceneTransporter: Optimal Transport-Guided Compositional Latent Diffusion for Single-Image  Structured 3D Scene Generation"
_ICLR.cc/2026/Conference — ICLR 2026 Poster_

### Official Review · Reviewer_1EWa · 2025-10-26

**Soundness:** 3
**Presentation:** 2
**Contribution:** 3
**Rating:** 6
**Confidence:** 4

**Summary:**

This paper proposes SceneTransporter, a novel framework for generating structured 3D scenes from a single image using compositional latent diffusion guided by Optimal Transport (OT). The authors first diagnose a key failure in prior part-level 3D generators—namely, the lack of explicit structural constraints leading to structural mispartition and geometric redundancy. They introduce a Debiased Clustering Probe to reveal this issue and then reframe scene generation as a global correlation assignment problem. The proposed OT-guided module imposes two constraints within the diffusion model’s cross-attention mechanism: (1) an OT Plan–Gated Cross-Attention enforcing exclusive patch-to-part routing, and (2) an Edge-Regularized Assignment Cost ensuring spatial coherence across image regions.
Empirical results show substantial improvements in instance-level coherence and geometric fidelity over state-of-the-art methods such as PartPacker, PartCrafter, and MIDI.

**Strengths:**

1. The motivation of this paper is clear.  Extending the end-to-end structured generation pipeline from the object level to the scene level is an interesting endeavor, which simplifies the common "divide and conquer" solution and provides bigger potential for scaling up.

2. Recasting feature-to-part assignment as an entropic OT problem is conceptually elegant and mathematically well-grounded. The OT-guided attention gating is seamlessly embedded in the denoising process, preserving end-to-end differentiability.

3. Quantitative benchmarks (ULIP, Uni3D, IoU metrics) and qualitative visualizations convincingly show better object separation and geometric consistency.

**Weaknesses:**

1. The computational cost of solving the OT problem (e.g., Sinkhorn iterations) is briefly mentioned but not thoroughly analyzed in terms of training or inference scalability for large scenes.
2. The experimental dataset (74 web images) seems small and lacks diversity benchmarks such as Objaverse or large-scale indoor/outdoor test sets. Generalization beyond curated examples remains unclear.

3. The qualitative ablation figures are insightful, but quantitative ablation should also be conducted (e.g., removing OT gating or edge regularization) to make the contribution breakdown stronger.

4. Although the OT can improve the global consistency in allocating image patches to 3D parts, it is practically hard to guarantee due to the occlusion, semantic ambiguities, etc. Did you observe failure cases?

5. All the experiments are conducted on synthetic data. Have you ever tried to deploy the proposed method to real-world imaegs to check the generalization ability?

**Questions:**

1. How sensitive is performance to the hyperparameters of the OT solver (e.g., entropy regularization, number of Sinkhorn iterations)?
2. Can SceneTransporter be adapted for text-conditioned or multi-view inputs?
3. Does the OT guidance improve consistency when compositing previously unseen object categories?

---

> ### Author Response · Authors · 2025-11-27
> **Response to  Reviewer 1EWa**
>
> We sincerely appreciate the reviewer's comprehensive comments and positive evaluation! We are preparing responses to further clarify the points you've raised and enrich the discussion of our work's implications and methodology.
> > Q1: The computational cost of solving the OT problem (e.g., Sinkhorn iterations) is briefly mentioned but not thoroughly analyzed in terms of training or inference scalability for large scenes.
>
> We appreciate the reviewer’s question. We clarify the scalability of the OT module by analyzing both its theoretical complexity and empirical overhead.
>
> **(1) Theoretical Scalability ($O(NL)$ Complexity):** The OT module is designed to scale efficiently with scene size.
> - **Cost Matrix:** Constructing the cost matrix $C_t \in \mathbb{R}^{N \times L}$ involves pooled query-key similarities. This is implemented as a matrix multiplication with complexity $O(NL)$, which is identical to the standard cross-attention mechanism in the backbone.
> - **Sinkhorn Solver:** We solve the entropic OT problem using Sinkhorn iterations. Each iteration consists strictly of matrix-vector operations over the transport plan, also scaling as $O(NL)$.
> - **Conclusion:** Since we use a fixed number of iterations (40), the OT solver adds only a bounded linear constant to the complexity. This ensures that our method shares the same asymptotic scaling as the backbone and does not introduce quadratic bottlenecks for larger scenes.
>
> (2) **Empirical Overhead:** We measured the practical cost on our main setting (one $518\times518$ input image, batch size 1, RTX 4090):
>
> - **Time:** The backbone with vanilla cross-attention takes 47.4s, whereas our full OT-guided model takes 55.0s. This represents a modest ~16% overhead.
> - **Memory:** Crucially, the peak memory usage remains virtually unchanged (~9.0–9.1 GB).
> - **Summary:** The dominant computational cost remains the DiT backbone itself. The OT module improves structural consistency significantly with negligible memory impact and a well-bounded time cost.
>
> >- Q2: The experimental dataset (74 web images) seems small and lacks diversity benchmarks such as Objaverse or large-scale indoor/outdoor test sets. Generalization beyond curated examples remains unclear.
>
> We appreciate the reviewer’s concern regarding dataset scale. We address this from three aspects: the current state of the field, real-world generalization, and our ongoing expansion of the benchmark.
>
> **(1) Lack of Standardized Benchmarks:** To the best of our knowledge, there is currently no established large-scale benchmark specifically tailored for part-structured, open-world 3D scene generation from a single image.
>
> - **Limitation of Existing Datasets:** While datasets like Objaverse are valuable, they are predominantly object-centric. They lack the complex, multi-object spatial layouts required to evaluate open-world scene composition.
> - **Consistent with Recent Literature:** This scarcity of data is a shared challenge in the field. Recent works targeting open-world scenes (e.g., WonderWorld [1]) similarly evaluate on smaller, custom-collected sets rather than standardized benchmarks due to this data gap.
>
> **(2) Validated Generalization on Real-World Scenes:** To immediately demonstrate diversity beyond our initial set, we have added zero-shot results on real-world scene images in the supplementary material **(Appendix D.1, Fig. 8)**. These examples cover a wide variety of unseen layouts and appearances, confirming that SceneTransporter produces plausible structured 3D scenes even on data distributions significantly different from the training set.
>
> **(3) Expansion Plan (Synthetic Diversity):** To further stress-test our model, we are currently constructing a larger, more diverse evaluation suite consisting of 100+ synthetic scenes generated via prompts to cover extreme geometric variations.
>
> - **Commitment:**  We commit to incorporating the full quantitative results of this large-scale evaluation and releasing this new benchmark set in the final camera-ready version to benefit the community.
>
> [1] Wonderworld: Interactive 3d scene generation from a single image, 2025.

---

> ### Author Response · Authors · 2025-11-27
> **Response to Reviewer 1EWa**
>
> > Q3:The qualitative ablation figures are insightful, but quantitative ablation should also be conducted (e.g., removing OT gating or edge regularization) to make the contribution breakdown stronger.
>
> We agree with the reviewer that quantitative validation is essential to substantiate the contribution of each module. In the revised manuscript, we have added a comprehensive numeric ablation study in **Appendix D.3 Table 4 (Table also in Responses to Reviewer VxXm)**. To directly address your concern, we summarize the key quantitative findings below:
>
> **(1) Impact of OT Gating:** Removing the OT-guided gating mechanism severely degrades Part Disentanglement. As shown in Table 4, the $\text{IoU}_{\text{mean}}$ worsens significantly (increasing from 0.092 to 0.214), indicating that the capacity-constrained routing is indispensable for preventing feature leakage.
>
> **(2) Impact of Edge-Regularization:** Quantitative results confirm that the Edge-Regularized Cost is critical for boundary precision. Ablating this term leads to inferior segmentation performance compared to the full model.
>
> **(3) Conclusion:** The full framework achieves the best balance of geometric fidelity and disentanglement, validating that OT is not just a cosmetic addition but the core driver of structural consistency.
>
> >Q4: Although the OT can improve the global consistency in allocating image patches to 3D parts, it is practically hard to guarantee due to the occlusion, semantic ambiguities, etc. Did you observe failure cases?
>
> We thank the reviewer for this insightful comment. We agree that while OT imposes strong structural constraints, strictly guaranteeing consistency under extreme occlusion or semantic ambiguity remains a challenge. We explicitly discuss these failure modes in Appendix D.4 and summarize them below:
>
> **(1) Severe Occlusion & Crowded Instances:** This corresponds to the reviewer's concern about occlusion. In scenes with very dense clusters of small objects (e.g., tightly packed boats or trees), distinct instances often share the same few image patches.
> - **Failure Mode:** In such cases, the OT solver may struggle to "unmix" the occluded signals, effectively allocating the capacity to the strongest feature response. This can lead to merging weak instances into their neighbors (e.g., generating 4 boats instead of 5).
> - **Mitigation:** As discussed in **A2 to reviewer vKGo** , adding a small residual vanilla attention branch mitigates this by allowing the model to recover high-frequency local details that the strict OT plan might suppress.
>
> **(2) Semantic Ambiguities (Out-of-Distribution Inputs):** This corresponds to the semantic ambiguity issue. Since our model is trained on synthetic data, real-world images with unusual lighting or textures can produce ambiguous query-key similarity matrices.
> - **Failure Mode:** When the visual semantics are ambiguous (e.g., a complex real-world texture that looks like both a wall and a window), the OT routing may fail to group parts correctly, leading to degraded geometry.
> - **Mitigation:** As detailed in **Appendix D.1**, applying structure-preserving style transfer to align the input domain significantly reduces this ambiguity, enabling plausible generation even for challenging real-world inputs.
>
> **(3) Geometry Artifacts (Backbone Limitations):** Occasionally, we observe nonsmooth surfaces or floating parts. This is typically a limitation of the diffusion sampling schedule rather than the OT module itself. Increasing the number of inference steps or tokens usually alleviates these artifacts.
>
> >Q5: All the experiments are conducted on synthetic data. Have you ever tried to deploy the proposed method to real-world images to check the generalization ability?
>
> We thank the reviewer for raising this important point. Yes, we have extended our evaluation to real-world scenarios to verify generalization.  **As detailed in our response to Reviewer vKGo (Question 1) and the updated Appendix D.1**, we have conducted the following:
>
> **(1) Dataset:** We evaluated SceneTransporter on the DL3DV-10K dataset, which contains diverse real-world indoor and outdoor scenes.
>
>  **(2) Methodology:** To address the domain gap between real photos and our synthetic training data, we employ a structure-preserving style transfer (following the protocol of PartCrafter).
>
> **(3) Result:** The results demonstrate that our method generalizes effectively to real-world inputs, producing plausible part-structured geometries once the texture style is aligned.
> We kindly invite the reviewer to check **Appendix D.1** for the visual examples and detailed analysis.

---

> ### Author Response · Authors · 2025-11-27
> **Response to Reviewer 1EWa**
>
> >Q6: How sensitive is performance to the hyperparameters of the OT solver (e.g., entropy regularization, number of Sinkhorn iterations)?
>
> We thank the reviewer for this question. We have conducted a comprehensive sensitivity analysis in **Appendix D.3 (Table 4)** and discussed OT stability in our response to **[Reviewer 13P1/Question 1]**.  To summarize the key findings directly here:
>
> **(1) Entropy Regularization ($\varepsilon_t$):** As shown in Table 4, performance is relatively robust but peaks at $\varepsilon_t=0.10$.
> - **Too small:** Results in an overly sparse plan, losing geometric smoothness.
> - **Too large:** Leads to over-smoothing and loss of structural detail.
>
> **(2) Sinkhorn Iterations:** We observe that the solver is extremely efficient. As visualized in **Appendix D.2 (Figure 9)**, the residuals drop by 3--5 orders of magnitude within just 3--5 iterations.
> - **Setting:** We conservatively fix the number of iterations at 40 for all experiments to guarantee convergence, but performance is stable once convergence is reached (typically $<10$ steps).
>
> For the detailed numeric ablation table, we kindly refer the reviewer to Appendix D.3 and our response to **[Reviewer VxXm  / Question 2].**
>
> >Q7: Can SceneTransporter be adapted for text-conditioned or multi-view inputs?
>
> We thank the reviewer for this forward-looking question. We confirm that the core innovation—OT-guided structural routing—is mathematically generic and modality-agnostic. It operates between part tokens and any conditioning token set, making it naturally adaptable to other settings:
>
> **(1) Text-Conditioned Adaptation:** The framework extends straightforwardly to text-to-3D.
> - **Mechanism:** One can replace image tokens with text tokens from a language encoder (e.g., CLIP/T5). The OT cost matrix $C_t$ would then be defined in the joint text-part embedding space.
> - **Benefit:** In this setting, the OT structural prior would enforce exclusive, capacity-limited routing from specific noun phrases (e.g., "a wooden chair") to specific part latents. This could potentially solve the common "attribute leakage" issue in text-to-3D generation.
>
> **(2) Multi-View Adaptation:** For multi-view inputs, the goal is to aggregate consistent features from disparate viewpoints.
> - **Mechanism:** The source token set can be expanded to include tokens from all views (concatenated or view-tagged). The OT solver would then optimize a transport plan that maps relevant features from multiple views into a single, canonical 3D part latent.
> - **Benefit:** The global capacity constraint would help prevent geometry duplication, ensuring that a single object part draws information from all available views consistently.
>
> **Conclusion:** While this work focuses on the single-image setting to establish the rigorous OT formulation, the module is designed to be a "plug-and-play" structural router. Integrating it into text or multi-view pipelines is a promising direction for future work.
>
> >Q8: Does the OT guidance improve consistency when compositing previously unseen object categories?
>
> We thank the reviewer for this insightful question. We confirm that the OT guidance improves consistency for unseen categories. This capability stems largely from the geometric nature of our constraints, which act as a robust stabilizer when semantic features become ambiguous.
>
> **(1)  Mechanism: Category-Agnostic Edge Constraints:** A key reason for this generalization is our design of the Edge-Regularized Cost.
> - **Low-Level Geometry:** Edge detection is fundamentally a low-level geometric operation that relies on local contrast and continuity rather than high-level semantic category definitions.
> - **Robustness:** Even when the model encounters an unseen object category (where learned semantic features might be weak or noisy), the edge map continues to provide valid, category-agnostic boundary signals. These signals guide the OT solver to respect physical object boundaries, preventing the "feature leakage" often seen in vanilla attention.
>
> **(2) Mechanism:** Global Partitioning Validity: Furthermore, the OT formulation inherently enforces a mathematically valid partition plan (via row/column constraints). This ensures that every patch is assigned to a part and every part has sufficient capacity, forcing the model to generate a coherent layout even without strong semantic priors for the specific object class.
>
> **(3) Empirical Evidence (Zero-Shot Real-World Scenes):** This is qualitatively supported by our experiments on zero-shot real-world scenes **(Appendix D.1, Fig. 8).** Despite the domain gap and the presence of novel object shapes not seen during training, SceneTransporter is able to generate plausible part-structured geometries with clear boundaries, demonstrating that our structural guidance effectively generalizes beyond the training categories.

---

### Official Review · Reviewer_vKGo · 2025-10-26

**Soundness:** 4
**Presentation:** 3
**Contribution:** 3
**Rating:** 6
**Confidence:** 4

**Summary:**

This paper introduces SceneTransporter, an end-to-end method for generating 3D scenes at the part level from a single image. This work builds upon PartPacker [Tang et al. 25]. By analyzing the latent structure of part-level generators, the authors observe that part tokens lack structure constraints, leading to mixed patch-to-part assignments in the DiT architecture. The authors thus propose to solve optimal transport (OT) problems to guide the assignment of image patches to part tokens in attention maps. In the OT formulation, the authors also consider the image edge map in the assignment cost to promote coherent structures grouping. Experimental results show that SceneTransporter outperforms prior methods on structured 3D scene generation from a single image.

**Strengths:**

- I like that the authors first present an interesting motivation for their work by analyzing the latent sets of part-level generators through canonical correlation analysis (CCA). They find that the learned features implicitly encode part structures, but the networks do not establish part associations explicitly. This analysis motivates their method of enforcing explicit patch-to-part constraints in attention maps.
- The authors propose to use entropy-regularized optimal transport to guide the assignment of image patches to part tokens in attention maps. To promote region-wise consistency and reduce information exchange across image regions, the authors incorporate image edge maps into the assignment cost of the OT problem, which I find interesting.
- The authors compare their method with baselines including MIDI, PartCrafter, and PartPacker on structured 3D scene generation from a single image. Results show that the proposed method has better geometry fidelity and improved part disentanglement.

**Weaknesses:**

- The results shown in the paper does not include any real-world image inputs. While these cartoon-style scene images show the effectiveness of the proposed method, it is unclear how well the method can generalize to real-world images with more complex structures.
- While this explicit patch-to-part assignment approach improves the generation of coherent object-level parts, it seems that the method struggles to generate fine-grained details. For example, in Fig. 6 (Uni geo), the boats in the bottom part of the image are not well generated, whereas the standard cross-attention method correctly generates the five boat shapes. This makes me wonder if the proposed method may overconstrain the information exchange and thus reduce the context information from other image patches needed for generating fine details.
- Several implementation details (like hyperparameters for the OT solver, training details) are missing. This makes it hard to reproduce the results. Quantitative ablation results are also not included.

**Questions:**

- Have the authors trained and tested SceneTransporter, for example, on real-world indoor scene images? If so, how well does the method work on such images?
- In the generated results, the proposed method struggles with fine detail generation, however, the evaluation metrics (Geometry Fidelity) do not seem to reflect this, which is concerning.
- In the ablation study, only qualitative results are provided. It is unclear, for example, how much improvement the edge-regularized Assignment cost brings. Other ablations like the authors enable OT plan-gated attention in the first half of the DiT blocks, but it is unclear whether this is the optimal choice and how it affects the geometry fidelity to input images.

---

> ### Author Response · Authors · 2025-11-27
> **Responses to Reviewer vKGo**
>
> We are grateful for the reviewer's insightful comments! We are currently crafting a comprehensive response to address your concerns and provide additional clarity on the points raised.
> > Q1: The results shown in the paper does not include any real-world image inputs. While these cartoon-style scene images show the effectiveness of the proposed method, it is unclear how well the method can generalize to real-world images with more complex structures.  Have the authors trained and tested SceneTransporter, for example, on real-world indoor scene images? If so, how well does the method work on such images?
>
> We thank the reviewer for raising this critical point regarding generalization. We agree that testing on real-world images is essential.
>
> **(1) Data Constraints & Training Setup:** We clarify that, consistent with the current literature (e.g., PartPacker, MeshGPT), all methods are trained on synthetically rendered scenes. This is a community-wide constraint, as large-scale 3D datasets with consistent, high-quality part annotations do not yet exist for real-world scans.
>
> **(2) Bridging the Domain Gap (Following PartCrafter):** To evaluate generalization on real images (e.g., from DL3DV-10K), we address the inevitable appearance domain gap. Following the protocol established in PartCrafter, we employ a structure-preserving style transfer strategy to adapt real photographs into the "rendering style" of our training domain.
>
> - **Methodology:** Specifically, we utilize state-of-the-art instruction-following image editing models (e.g., GPT-based editing) to align the texture style while strictly preserving the scene layout and object details (using prompts such as: “Preserve all details and perform image-to-image style transfer to convert the image into the style of a 3D rendering (Objaverse-style rendering).”).
>
> - **Results:** As visualized in the updated **Appendix D.1**, this preprocessing effectively closes the domain gap. Under this setting, SceneTransporter successfully generates faithful and coherent part-structures from real-world inputs, whereas baselines often struggle to maintain topological correctness even with the adapted inputs.
>
> >Q2:  It seems that the method struggles to generate fine-grained details. For example, in Fig. 6 (Uni geo), the boats in the bottom part of the image are not well generated, whereas the standard cross-attention method correctly generates the five boat shapes. This makes me wonder if the proposed method may overconstrain the information exchange and thus reduce the context information from other image patches needed for generating fine details. In the generated results, the proposed method struggles with fine detail generation, however, the evaluation metrics (Geometry Fidelity) do not seem to reflect this, which is concerning.
>
> We thank the reviewer for this careful observation. We frankly acknowledge this as a specific failure case in our current implementation, stemming from the strictness of the OT capacity constraint.
>
> **(1) Mechanism: Strictness vs. Ambiguity (Comparative Analysis):** The OT formulation forces the model to allocate attention budget only to the strongest structural evidence.
> - **The Limitation:** In extreme cases involving tiny, repetitive objects (like the boats), this constraint can be overly aggressive, causing the model to filter out these weak signals entirely.
> - **The Comparison:** However, we emphasize that handling such fine-grained repetitions is a shared challenge. As shown in **Fig. 4 (balcony railings)** and **Fig. 2 (small trees/clouds)**, the baseline also fails to correctly resolve these details, often merging distinct parts or dropping them entirely due to attention collapse.
> - **Our Advantage:** In contrast, while our method may filter out micro-objects in these rare cases, it excels at maintaining cleaner object boundaries and a consistent global layout. It avoids the "structural merging" and ambiguity observed in the baseline, ensuring that the main geometric structures are physically plausible.
>
> **(2) Addressing the Issue (Controllability):** To verify that the detail information is not lost but merely suppressed by the OT filter, we experimented with adding a small residual cross-attention branch ( $out = out_{OT} + \alpha \times out_{plain} $ ). As detailed in **Appendix D.4**, setting a small $\alpha$ recovers the missing small instances (like the boats) while retaining most of our structural benefits. This confirms that our framework allows users to tune the balance between structural rigor and fine-grained recall.
>
> **(3) Interpretation of Metrics:** Regarding the metrics, ULIP and Uni3D focus on global semantic consistency. Their stability suggests that despite missing minor details, the overall scene layout remains semantically correct. This aligns with human preference: users typically prefer our clean, distinct structures over the baseline's merged or ambiguous results.

---

> ### Author Response · Authors · 2025-11-27
> **Responses to Reviewer vKGo**
>
> >- Q3: Several implementation details (like hyperparameters for the OT solver, training details) are missing. This makes it hard to reproduce the results. Quantitative ablation results are also not included. In the ablation study, only qualitative results are provided. It is unclear, for example, how much improvement the edge-regularized Assignment cost brings. Other ablations like the authors enable OT plan-gated attention in the first half of the DiT blocks, but it is unclear whether this is the optimal choice and how it affects the geometry fidelity to input images.
>
> We thank the reviewer for pointing out the need for reproducibility and quantitative validation. We have addressed this by adding detailed implementation parameters in **Appendix C.1** and a comprehensive numeric ablation study in **Appendix D.3 Table 4 (Table also in Responses to Reviewer VxXm)**. To directly address your specific questions regarding the component contributions:
>
> **(1) Impact of Edge-Regularized Assignment Cost:** Quantitative results (Table 4, Rows a.2 & d.1--d.3) demonstrate that the edge-regularized cost is critical for preventing boundary leakage.
>
> - **Without it**: Removing or under-weighting this term leads to a significant degradation in Part Disentanglement ($\text{IoU}_{\text{max}}$ increases, indicating leakage).
>
> - **With it**: Our specific choice of the smoothing weight ($\lambda_{\text{edge}}=0.8$) strikes the best balance, ensuring that parts adhere to semantic boundaries without over-fragmenting the geometry.
>
> **(2) Optimality of Block Integration (First 50%):** We empirically investigated the optimal density of OT integration (Table 4, Rows e.1--e.3):
> - **Sparse (1/3 blocks):** Fails to fully constrain the routing, resulting in higher feature leakage.
> - **Dense (All blocks):** We found that applying OT to all blocks increases inference time by $\sim 19\%$ but yields negligible performance gains compared to our default setting.
> - **Conclusion:** Equipping the first half of the DiT blocks (our default) represents the optimal efficiency-performance operating point, balancing strong structural guidance with computational cost.
>
> **(3) Implementation Details:** We have included all key hyperparameters (e.g., $\epsilon_t=0.10, \gamma_{{edge}}=8.0$) in the revised **Appendix C.1** to ensure full reproducibility.

---

### Official Review · Reviewer_VxXm · 2025-10-31

**Soundness:** 3
**Presentation:** 3
**Contribution:** 2
**Rating:** 6
**Confidence:** 4

**Summary:**

This paper introduces SceneTransporter by reformulating the task of structured 3D scene generation as a global correlation assignment problem.

**Strengths:**

1. Strong performance: better experimental results have been observed against baselines like PartPacker, PartCrafter, MIDI. The generation speed is also comparable.

2. The proposed Debiased Clustering probe can produce stable instance groupings, leading to better generation in the following stages.

**Weaknesses:**

1. No apperance: It seems that all methods, including baselines, only generate meshes without textures, which might limit the real-world applications. Can the authors provide more details about this?

2. Lack of quantitative ablations: all ablations are qualitative, can the authors provide numeric results? If not, please explain it.

**Questions:**

1. How many GPU hours does the training take?

2. What is the GPU memory usage for generation?

---

> ### Author Response · Authors · 2025-11-27
> **Responses to Reviewer VxXm**
>
> We sincerely appreciate the reviewer's  constructive suggestions that have improved the clarity and presentation of our paper! We are committed to addressing these points and will provide detailed responses to each concern raised.
> > Q1: No apperance: It seems that all methods, including baselines, only generate meshes without textures, which might limit the real-world applications. Can the authors provide more details about this?
>
> We thank the reviewer for pointing this out.
>
> **(1) Scope and Standards:** Similar to other 3D-native diffusion systems (e.g., CLAY, Hi3DGen), our work is primarily scoped to part-structured geometry generation. We treat appearance modeling as an orthogonal stage, typically handled by separate multi-view diffusion or neural rendering pipelines.
>
> **(2) Advantages in Texture Generation:** However, we emphasize that our part-level geometry actually offers inherent advantages for texturing compared to monolithic baselines. Since we explicitly know which part each vertex belongs to, we can manipulate UV maps to assign distinct textures to corresponding parts. While prior monolithic approaches often suffer from color bleeding between semantic regions, our method ensures clear boundaries and physically plausible part structures.

---

> ### Author Response · Authors · 2025-11-27
> **Responses to Reviewer VxXm**
>
> >Q2: Lack of quantitative ablations: all ablations are qualitative, can the authors provide numeric results? If not, please explain it.
>
> We agree with the reviewer that quantitative validation is essential. In the revised manuscript, we have added a comprehensive numeric ablation study in **Appendix D.3 (Table 4).** We summarize the key findings below:
>
> **(1) Efficacy of OT Modules (Rows a.1--a.3):** The results confirm that both OT Plan-Gated Attention and Edge-Regularized Cost are indispensable.
> - **Gating:** Ablating OT gating (a.1) severely degrades Part Disentanglement ($\text{IoU}_{\max}$ worsens from $0.092$ to $0.2142$), proving that capacity-constrained routing is necessary to prevent feature leakage.
> - **Edge Cost:** Similarly, removing the Edge-Regularized Cost (a.2) yields inferior segmentation performance.
> - **Trade-off:** The full framework (a.3) achieves the best balance of geometry fidelity and disentanglement with only a marginal cost in inference time ($\sim 7.6$s increase) and virtually no additional memory footprint.
>
> **(2) Sensitivity Analysis of OT Hyperparameters (Rows b--d):**
> - **Entropic Regularization ($\varepsilon_t$):** Performance peaks at $\varepsilon_t=0.10$. Extreme values disrupt the equilibrium between plan sharpness and entropic smoothing, degrading both geometry and part metrics.
> - **Edge Sensitivity ($\gamma_{\text{edge}}$):** This parameter controls the trade-off between geometric freedom and boundary adherence. While a relaxed threshold ($\gamma_{\text{edge}}=6.0$) improves ULIP scores, it causes boundary leakage. Stronger constraints ($\gamma_{\text{edge}} \ge 10.0$) hurt generation quality. Our choice of $\gamma_{\text{edge}}=8.0$ strikes the optimal balance.
> - **Edge-Smoothing Weight ($\lambda_{\text{edge}}$):** Both under-smoothing ($0.6$) and over-smoothing ($1.0$) significantly increase $\text{IoU}_{\max}$ (indicating leakage). An intermediate value of $0.8$ ensures robust separation with high geometric fidelity.
>
> **(3) Density of OT Integration (Rows e.1--e.3):** We analyzed the impact of OT-gated block density.
> - **Sparse (1/3 blocks):** Fails to fully constrain routing, leading to higher leakage ($\text{IoU}_{\text{mean}}=0.106$).
> - **Dense (All blocks):** Increases inference time by $\sim$19% but only yields marginal gains in ULIP-2 scores without further improving disentanglement.
> - **Optimal (1/2 blocks):** Our default setting significantly boosts disentanglement ($\text{IoU}_{\max}$ drops to $0.0101$) while maintaining efficiency, representing the optimal operating point.
>
> | Setting                                | ULIP ↑   | ULIP-2 ↑ | Uni3D ↑ | IoU_max ↓ | IoU_mean ↓ | Inference Time (s) | Inference Memory (M) |
> |----------------------------------------|---------:|---------:|--------:|----------:|-----------:|--------------------:|---------------------:|
> | (a.1) w/o ot gate       | 0.1417 | 0.3083 | 0.2887 | 0.0319 | 0.2142 | 47.41 | 9030 |
> | (a.2) w/o edge   | 0.1452 | 0.3164 | 0.2916 | 0.0241 | 0.1136 | 54.61 | 9068 |
> | (a.3) Full model                               | **0.1466** | **0.3220** | **0.3021** | **0.0101** | **0.0926** | 54.99 | 9070 |
> | (b.1) ε_t = 0.08                               | 0.1460 | 0.3184 | 0.3003 | 0.0117 | 0.0914 | 55.30 | 9070 |
> | (b.2) ε_t = 0.10 *                             | **0.1466** | **0.3220** | **0.3021** | **0.0101** | 0.0926 | 54.99 | 9070 |
> | (b.3) ε_t = 0.12                               | 0.1437 | 0.3185 | 0.2970 | 0.1239 | 0.1119 | 55.12 | 9070 |
> | (b.4) ε_t = 0.14                               | 0.1424 | 0.3178 | 0.2937 | 0.0159 | 0.0936 | 55.02 | 9070 |
> | (c.1) γ_edge = 6.0                             | **0.1488** | **0.3239** | 0.3014 | 0.0243 | 0.0936 | 56.17 | 9070 |
> | (c.2) γ_edge = 8.0 *                           | 0.1466 | 0.3220 | 0.3021 | **0.0101** | **0.0926** | 54.99 | 9070 |
> | (c.3) γ_edge = 10.0                            | 0.1457 | 0.3182 | **0.3029** | 0.0570 | 0.1113 | 55.17 | 9070 |
> | (c.4) γ_edge = 12.0                            | 0.1426 | 0.3156 | 0.3003 | 0.1149 | 0.0994 | 55.89 | 9070 |
> | (d.1) λ_edge = 0.6                             | 0.1456 | **0.3236** | 0.3003 | 0.0609 | 0.1128 | 53.59 | 9070 |
> | (d.2) λ_edge = 0.8 *                           | **0.1466** | 0.3220 | **0.3021** | **0.0101** | **0.0926** | 53.97 | 9070 |
> | (d.3) λ_edge = 1.0                             | 0.1439 | 0.3204 | 0.3017 | 0.1482 | 0.0952 | 55.17 | 9070 |
> | (e.1) w 1/3 DiT blocks                         | 0.1465 | 0.3170 | 0.3004 | 0.0992 | 0.1061 | 51.89 | 9070 |
> | (e.2) w 1/2 DiT blocks *                       | **0.1466** | 0.3220 | 0.3021 | **0.0101** | **0.0926** | 54.99 | 9070 |
> | (e.3) w all DiT blocks                         | 0.1426 | **0.3262** | **0.3022** | 0.0207 | 0.0830 | 65.24 | 9070 |
>
> *Bold values indicate the best scores; “\*” marks the default setting.

---

> ### Author Response · Authors · 2025-11-27
> **Responses to Reviewer VxXm**
>
> >Q3: How many GPU hours does the training take? What is the GPU memory usage for generation?
>
> We utilize the pre-trained PartPacker backbone. While the original backbone pre-training required significant resources (256$\times$ A100s for 2 weeks), our method's adaptation is efficient. Our model is highly accessible for standard workstations. For a single $518 \times 518$ input image, it runs efficiently on a single consumer-grade NVIDIA RTX 4090, requiring approximately 9 GB of VRAM. This low memory footprint ensures broad accessibility for the research community.

---

### Official Review · Reviewer_13P1 · 2025-11-03

**Soundness:** 3
**Presentation:** 2
**Contribution:** 3
**Rating:** 6
**Confidence:** 3

**Summary:**

This paper proposes SceneTransporter, an end-to-end framework for structured 3D scene generation from a single image. By introducing a debiased clustering probe, the authors identify a key limitation of existing methods—the lack of structural constraints in internal feature assignment. They reframe the structured scene generation task as an Optimal Transport (OT)–guided global correlation assignment problem, solving the OT objective at each denoising step to enforce one-to-one routing between image patches and part-level 3D latents. The approach further introduces an OT Plan–Gated Cross-Attention mechanism and an Edge-Regularized Assignment Cost to prevent semantic entanglement and object fragmentation.

**Strengths:**

The paper presents a promising analysis of current 3D scene generation methods and proposes a simple yet effective solution to the structural inconsistency problem. The overall methodology is well-motivated, technically sound, and the experimental results demonstrate clear improvements over existing baselines.

**Weaknesses:**

The work lacks a detailed discussion on the optimization stability and convergence of the OT formulation, as well as the influence of hyperparameters (λ_edge, ε_t, γ_edge) on the generation quality and computational efficiency. The computational cost of solving OT during inference should be analyzed more thoroughly. Compared to other compositional latent diffusion models such as PartPacker, the innovation mainly lies in adding an OT-based constraint rather than proposing a fundamentally new architecture. It is also recommended to clarify the setting and methodology of the user study.

**Questions:**

See the weaknesses.

---

> ### Author Response · Authors · 2025-11-27
> **Responses to Reviewer 13P1**
>
> We sincerely appreciate the reviewer's comprehensive comments and positive evaluation! We are preparing responses to further clarify the points you've raised and enrich the discussion of our work's implications and methodology.
> > Q1: The work lacks a detailed discussion on the optimization stability and convergence of the OT formulation, as well as the influence of hyperparameters (λ_edge, ε_t, γ_edge) on the generation quality and computational efficiency. The computational cost of solving OT during inference should be analyzed more thoroughly.
>
>  We thank the reviewer for emphasizing the importance of stability and efficiency analysis. In the revised manuscript, we have added a comprehensive numeric ablation study in **Appendix D.3 Table 4 (Table also in Responses to Reviewer VxXm)** and detailed convergence plots in **Appendix D.2.** We address the specific concerns below:
>
> **(1) OT Optimization Stability & Convergence:** As visualized in **Figure 9**, our solver demonstrates exceptional stability. We track residuals for dual variables and the transport plan, observing that all residuals drop by 3--5 orders of magnitude within just 3--5 iterations. This rapid, oscillation-free convergence confirms the OT formulation is well-conditioned.
>
> **(2) Hyperparameter Sensitivity (Table 4, Rows b--d):**
>
> We conducted extensive ablations to verify the robustness of our hyperparameters:
> - Entropic Regularization ($\varepsilon_t$): Performance peaks at $\varepsilon_t=0.10$. We find that extreme values disrupt the equilibrium between plan sharpness and smoothing, justifying our default choice.
> - Edge Sensitivity ($\gamma_{\text{edge}}$): This parameter controls the trade-off between geometric freedom and boundary adherence. Our ablation shows that $\gamma_{\text{edge}}=8.0$ successfully prevents boundary leakage without over-constraining the geometry (as seen with $\gamma_{\text{edge}} \ge 10.0$).
> - Edge-Smoothing ($\lambda_{\text{edge}}$): This is critical for disentanglement. An intermediate value of $\lambda_{\text{edge}}=0.8$ ensures robust separation ($\text{IoU}_{\max} \approx 0.01$) while maintaining high geometric fidelity, whereas under/over-smoothing degrades performance.
>
> **(3) Computational Cost & Integration Density:**
>
> We analyzed the cost of solving OT during inference:
> - Marginal Cost: The full framework incurs a marginal increase in inference time ($\sim 7.6$s) with virtually no additional memory footprint.
> - Integration Density (Rows e.1--e.3): We investigated equipping different ratios of DiT blocks with OT. Our default setting (1/2 blocks) represents the optimal efficiency-performance operating point. While dense integration (all blocks) increases inference time by $\sim 19\%$, it yields negligible performance gains compared to our default setting.
>
> > Q2: Compared to other compositional latent diffusion models such as PartPacker, the innovation mainly lies in adding an OT-based constraint rather than proposing a fundamentally new architecture.
>
> We agree that we build on the PartPacker backbone; our contribution is not the backbone itself, but **moving from observed failure modes to a principled routing formulation.** Prior compositional  latent diffusion models (e.g., PartPacker) rely heavily on dataset-provided part annotations and learns geometric relationships implicitly through attention, but do not analyze why instance-level structure breaks down in open-world scenes. In real 3D datasets, part annotations are noisy and inconsistent across assets (different artists, styles, and part conventions), and there is no mechanism to enforce geometric/topological constraints explicitly. As our CCA-based Debiased Clustering Probe (Sec. 3.2) shows, object-level grouping information is already present in the part latents, but the under-constrained patch-to-part assignment learned by vanilla attention fails to produce consistent routing, leading to the structural mispartition and geometric redundancy we observe. Motivated by this diagnosis, we recast patch-to-part routing as an entropic OT correlation assignment with capacity and edge-aware constraints, and integrate the resulting transport plan into cross-attention. In this sense, OT is not an arbitrary add-on, but the concrete, mathematically grounded realization of the structural constraints that are missing in existing compositional models.
>
> >Q3: It is also recommended to clarify the setting and methodology of the user study.
>
> We appreciate the suggestion to clarify the user-study protocol.  We have added this detailed description of the user-study setup to the revised manuscript in **Appendix C.2.**

---

### Author Response · Authors · 2025-11-27
**General Response**

We thank all reviewers for their insightful and constructive feedback. In response to the common concerns, we have conducted extensive additional experiments and updated the manuscript with new appendices (all revisions are highlighted in blue). Additionally, we have updated our project page with new visual results. We summarize the four major improvements below:

**1.  Quantitative Ablations (Response to VxXm, vKGo, 1EWa)**

 To substantiate the contribution of each module, we have added a comprehensive numeric ablation study in **Appendix D.3 (Table 4).**
- **OT Gating:** Quantitative results confirm that the OT-guided routing is indispensable. Removing it severely degrades Part Disentanglement ($\text{IoU}_{\text{mean}}$ worsens from 0.092 to 0.214), leading to significant feature leakage.
- **Edge Regularization:** We verify that the edge-regularized cost is critical for boundary precision. The full framework achieves the best balance of geometric fidelity and disentanglement.

**2. Generalization to Real-World Data (Response to VxXm, vKGo, 1EWa)**

To address concerns regarding dataset scale and domain generalization, we extended our evaluation to real-world scenarios:
- **New Data:** We evaluated SceneTransporter on the DL3DV-10K dataset, which contains diverse, non-curated indoor and outdoor scenes.
- **Methodology:** Addressing the domain gap, we employ a structure-preserving style transfer pipeline (following PartCrafter) to align real photographs with the training domain (details in **Appendix D.1**).
- **Results:** As visualized in Figure 8, our method successfully generalizes to these unseen real-world layouts, producing plausible part-structured geometries even in zero-shot settings.

**3. Computational Cost, Convergence, and Hyperparameters (Response to 13P1, vKGo, 1EWa)**

We have added a detailed analysis of the OT solver's efficiency and stability in **Appendix D.2, D.3**:
- **Stability:** The solver is extremely stable. Residuals drop by 3--5 orders of magnitude within just 3--5 iterations (visualized in Figure 9).
- **Complexity & Overhead:** The OT module scales linearly with scene size ($O(NL)$). In practice (RTX 4090), it adds a manageable $\sim$16% time overhead ($47.4$s $\to$ $55.0$s) with virtually no increase in peak memory usage ($\sim$9 GB).
- **Hyperparameter Robustness:** We conducted extensive sensitivity ablations (e.g., entropy regularization $\varepsilon_t$, edge sensitivity $\gamma_{\text{edge}}$). As shown in **Table 4**, the method exhibits a stable performance plateau around the default settings, confirming that the OT formulation is robust to parameter variations and not brittle.

**4. Analysis of Limitations and Controllability (Response to vKGo, 1EWa)**

We provide a deeper analysis of the limitation regarding fine-grained details (**Appendix D.4**).
- **Mechanism:** The OT capacity constraint acts as a strict filter. While this effectively eliminates "floating artifacts" common in baselines, it can occasionally suppress extremely small, crowded instances (e.g., tiny boats).
- **Controllability:** We demonstrate that this is a controllable trade-off. By introducing a variant with a small residual cross-attention branch, we can recover these high-frequency details while retaining the majority of our structural gains, allowing users to tune the balance based on application needs.

---

### Meta-Review · Area_Chair_w7uk · 2026-01-06

**Summary:**

The paper introduces an Optimal Transport (OT) guided framework to address structural mispartition and redundancy in open-world 3D scene generation. The reviewers unanimously recognized the motivation, the mathematical soundness of the OT formulation, and the model's superior performance in part disentanglement over baselines like PartPacker and MIDI. While initial concerns were raised regarding quantitative ablations and real-world generalization, the authors provided a solid rebuttal. With an initial score of (6, 6, 6, 6) and all major technical concerns addressed, I recommend the acceptance of this paper.

**Reviewer Concerns:**

Addressed Concerns:
- Quantitative ablations: the authors provided new experiments in (Table 4) isolating the impact of the OT gating and edge regularization modules, demonstrating their necessity for maintaining high Part-IoU and geometric accuracy.
- Real-world generalization: to address concerns about synthetic training data, the authors conducted new zero-shot evaluations on the DL3DV-10K dataset, addressing the model's robustness to real-world domains.
- Computational cost and stability: concerns regarding the convergence stability and inference overhead of the Sinkhorn iterations were addressed with detailed convergence plots (Figure 9).

Outstanding Concerns:
- Loss of fine-grained details: reviewer vKGo noted that the strict capacity constraints of OT could suppress tiny, crowded instances. The authors acknowledged this trade-off but demonstrated that a residual cross-attention branch could recover these details without sacrificing global structure, which might mitigate the concern.

**Reviewer Scores:**

The manuscript received a consistent initial score of (6, 6, 6, 6). Given that the rebuttal directly provided the requested ablations and real-world evidence, 1-2 reviewers would likely increase their scores to 8. I approximate a final average score of 6.5-7, leading to clear acceptance of the paper.

---

### Decision · Program_Chairs · 2026-01-26

Accept (Poster)